# FlashResearch: Real-time Agent Orchestration for Efficient Deep Research

## Abstract

Deep research agents, which synthesize information across diverse sources, are significantly constrained by their sequential reasoning processes. This architectural bottleneck results in high latency, poor runtime adaptability, and inefficient resource allocation, making them impractical for interactive applications. To overcome this, we introduce **FlashResearch**, a novel framework for efficient deep research that transforms sequential processing into parallel, runtime orchestration by dynamically decomposing complex queries into tree-structured sub-tasks. Our core contributions are threefold: **(1)** an **adaptive planner** that dynamically allocates computational resources by determining research breadth and depth based on query complexity; **(2)** a **real-time orchestration layer** that monitors research progress and prunes redundant paths to reallocate resources and optimize efficiency; and **(3)** a **multi-dimensional parallelization framework** that enables concurrency across both research breadth and depth. Experiments show that FlashResearch consistently improves final report quality within fixed time budgets, and can deliver up to a $5\times$ speedup while maintaining comparable quality.

## 1 Introduction

Deep research tasks, which involve synthesizing information from diverse sources and navigating complex, interdependent concepts, pose significant challenges for existing AI systems. These tasks often demand knowledge retrieval, advanced reasoning, sophisticated tool use, and dynamic planning over multiple steps under structural uncertainty and evolving objectives (Du et al., 2025). Applications include literature review (Haman & Školník, 2025), open-domain question answering, and policy analysis (Gambrell, 2025), where the ability to evaluate conflicting perspectives, explore hypotheses, and revise beliefs as new evidence emerges is essential. However, current systems often take tens of minutes to respond. This latency can break users' cognitive flow (Iqbal & Horvitz, 2007), incur high context-switching costs (Mark et al., 2008), and degrade overall experience.

Much of this inefficiency stems from poor orchestration. Existing systems typically rely on sequential processing (Xu & Peng, 2025), leading to unnecessary latency when subproblems are independent, leaving parallelizable evidence gathering, hypothesis branching, and speculative exploration underexploited. Moreover, static planning strategies in these systems also fail to adapt to the dynamic nature of research (Zheng et al., 2025b). The value of subqueries or deeper investigation often becomes clear only during execution, but current systems rarely prune low-value paths or reallocate effort during runtime.

To address these challenges, we propose **FlashResearch**, a framework designed for efficient deep research that integrates adaptive planning, real-time orchestration, and concurrent execution. FlashResearch treats deep research as a dynamic, tree-structured traversal, where a complex query is decomposed into concurrent subqueries that dynamically populate the research tree. The objective is to maximize response quality under a time budget by adjusting the tree structure and reallocating effort across promising research directions.

At the core of FlashResearch is an adaptive planner that decides, at each step, how many subqueries to open and whether to explore further depth, based on the complexity of the input query. It weighs the expected marginal utility of each branch, expanding breadth when broad exploration is valuable and deepening paths selectively when the information gain justifies further effort. This allows

FlashResearch to flexibly allocate resources depending on whether the query is on a broad topic that demands diverse perspectives, or a specific one that requires deeper investigation.

However, planning alone is insufficient. Since research is inherently iterative and non-linear, newly emerged evidence may reshape priorities mid-execution. FlashResearch incorporates a real-time orchestration layer that monitors ongoing research outputs, evaluates them against goal satisfaction and quality metrics, and makes live adjustments. This mechanism allows the system to terminate low-value or redundant branches early and reallocate computational resources toward more promising paths. More importantly, it allows speculative execution by launching child tasks before parent-level planning decisions are finalized, thereby reducing idle time and accelerating throughput.

This tight feedback loop between planning and execution produces a highly dynamic research tree that evolves in response to both query structure and emergent information. To handle this, FlashResearch employs a multi-dimensional parallelization framework to schedule tasks across breadth and depth through a unified task pool. Built on a fully asynchronous infrastructure with thread-safe state management, the system enables simultaneous exploration of multiple research paths and allows non-blocking orchestration to adapt the tree structure as new findings emerge.

To evaluate the effectiveness of FlashResearch, we conduct experiments on two recent deep research benchmarks, DeepResearchGym and DeepResearch Bench (Coelho et al., 2025; Alzubi et al., 2025). Compared to the baseline, FlashResearch can consistently accomplish deeper and wider research within fixed time constraints, producing research reports of better comprehensiveness and insights.

The key contributions of this work are:

- A formal formalization of deep research tasks as a tree-structured optimization problem.
- An adaptive planning module for real-time, context-aware decisions on task branching, recursion, and termination.
- A real-time orchestration framework for dynamic task monitoring, speculative execution, and intelligent resource reallocation.
- A fully asynchronous and parallelized execution architecture enabling concurrent research execution across multiple dimensions.
- A comprehensive empirical evaluation demonstrating FlashResearch's superior improvements in terms of throughput, quality, and efficiency on complex research tasks.

## 2 RELATED WORKS

### 2.1 DEEP RESEARCH AGENTS

Building on earlier tool-use frameworks like WebGPT (Nakano et al., 2021) and ReAct (Yao et al., 2023b), recent deep research agents – including GPT-Researcher (Elovic, 2023), Open Deep Search (Alzubi et al., 2025), and LangChain's Open Deep Research (Langchain, 2025) – decompose complex queries into tool-augmented subtasks. To standardize evaluation, emerging benchmarks like DeepResearchGym (Coelho et al., 2025) and DeepResearch Bench (Du et al., 2025) introduce LLM-as-a-judge protocols tailored for complex research questions.

Despite these advances, current systems typically rely on fixed, pre-specified parameters for controlling the research structure. Their orchestration strategies are dominated by sequential execution or coarse-grained parallelism (Xu & Peng, 2025), which limits adaptability. As a result, when information quality shifts during execution, these systems either waste compute or incur unnecessary latency. FlashResearch addresses these limitations by introducing a real-time orchestration layer that couples multi-dimensional parallelism across both depth and breadth. Unlike prior static approaches, it adaptively expands or prunes subqueries in real time based on intermediate evidence, enabling more efficient and responsive deep research.

### 2.2 AGENTIC WORKFLOW ORCHESTRATION

Recent work compiles high-level goals into executable agent graphs via MCTS-guided code search (Zhang et al., 2024), evolutionary populations of heterogeneous workflows (Niu et al., 2025), and modular activity-on-vertex graphs (Zhang et al., 2025a). These systems mainly optimize *offline* and then execute largely fixed graphs, and their runtime control over partially executed graphs remains

Figure 1: Overview of FlashResearch: the Planning Nodes adaptively decompose queries into parallel subqueries executed by Research Nodes for findings, which may recursively trigger deeper planning. The **adaptive planner** expands (1) *breadth* to explore prior-research and regulates (2) *depth* to pursue promising paths post-research. The **real-time orchestration layer** monitors progress and reallocates resources through (3) *scheduling signals* mid-research. A **multi-dimensional parallelization framework** enables flexible concurrency across both breadth and depth.

limited. Production frameworks such as AutoGen (Wu et al., 2023), LangGraph (LangChain, 2024), DSPy (Khattab et al., 2024), and OpenAI Swarm (OpenAI, 2024) provide valuable abstractions for multi-agent pipeline control.

More recently, hierarchical frameworks like OWL (Hu et al., 2025) deploy domain-agnostic planners with specialized workers using task decomposition and reactive adaptive, while AgentOrchestra (Zhang et al., 2025c) unifies tools, environments, and agents via TEA Protocol for agent-level coordination. Cognitive Kernel-Pro (Fang et al., 2025) focuses on cognitive architecture and memory for knowledge integration, and OAgents (Zhu et al., 2025) provides a modular infrastructure for agent design and evaluation. For runtime adaptation, Co-Sight (Zhang et al., 2025b) embeds replanning rules in its agentic planner, smolagents (Roucher et al., 2025) employs interval-based replanning up to fixed step limits, and dynamic workflow systems (Gao et al., 2025; Nie et al., 2025) enable intelligent agent resource allocation. However, these approaches may struggle with the unique demands of deep research tasks where tree-structured exploration management is important for balancing multiple parallel branches and adapting strategies based on intermediate findings under time constraints.

FlashResearch addresses these gaps through *finer-grained real-time* orchestration with task monitoring, dynamic resource reallocation, and mid-execution suspension, escalation, and replanning.

## 2.3 PARALLEL AND SPECULATIVE REASONING

Token- and action-level acceleration methods such as speculative decoding (Leviathan et al., 2023; Miao et al., 2023; Cai et al., 2024) and speculative reasoning for fast inference (Pan et al., 2025; Yang et al., 2025) reduce latency via draft-and-verify or multi-token prediction. At the reasoning level, Dynamic Parallel Tree Search (Ding et al., 2025) accelerates Tree-of-Thoughts by expanding and pruning nodes in parallel, while ParaThinker (Wen et al., 2025) and Parallel-R1 (Zheng et al., 2025a) instill native parallel reasoning. These improve efficiency and accuracy but still rely on static branching. Inspired by these works, FlashResearch advances parallelism to the workflow level: it not only reallocates compute and prunes branches dynamically, but also supports speculative execution—allowing branches to expand without delay and later discarding them if evidence shows they are unnecessary.

## 3 BACKGROUND

### 3.1 FORMULATING DEEP RESEARCH

Deep research tasks involve tackling complex, open-ended queries by multi-step reasoning, gathering diverse information, and synthesizing knowledge into comprehensive responses. We formalize such a task as follows: a user query $q \in \mathcal{Q}$, where $\mathcal{Q}$ is the space of natural language queries. The goal is to produce a response $r \in \mathcal{R}$ by integrating knowledge from retrieved context $C = \{c_1, c_2, \ldots, c_n\}$ sourced from a corpus $D$ (e.g., web searches or local documents). During this process, research findings $F = \{f_1, f_2, \ldots, f_m\}$, comprising reasoning artifacts and key insights, are iteratively derived from the context. Consequently, the response is generated as

$$r = \sigma(q, C, F), \tag{1}$$

where $\sigma : \mathcal{Q} \times 2^C \times 2^F \to \mathcal{R}$ is a synthesis function that aggregates and refines the inputs to maximize factual accuracy, comprehensiveness, and relevance. Here, $2^C$ and $2^F$ denote the power sets of $C$ and $F$, representing all possible subsets of contexts and findings, respectively. In practice, these are subsets selected based on relevance constraints, and $\sigma$ is typically realized by an LLM agent.

To solve such tasks scalably, a deep research framework must balance the thoroughness of exploration with computational overheads. Conventional sequential pipelines—such as linear chains of retrieval-augmented generation (RAG) steps—often falter under intricate queries, incurring high costs from redundant traversals or premature convergence to suboptimal paths. Given the hierarchical and multi-faceted nature of deep research, it is natural to model the process as a tree structure, like in Tree of Thoughts (Yao et al., 2023a).

Formally, we model the process as a directed tree $\mathcal{T} = (N^P \cup N^R, E)$ with disjoint node sets of *planning nodes* $N^P$ and *research nodes* $N^R$.

A planning node $n^P$ decomposes a query $q^n$ into a finite set of subqueries:

$$n^P(q^n) \to \{q_1^n, \ldots, q_{b_n}^n\}, \quad q_j^n \in \mathcal{Q}. \tag{2}$$

Here, $q^n$ can be either the initial query or a subquery generated by a previous planning node. Each $q_j^n \in n^P(q^n)$ instantiates a research node $n^R(q_j^n)$. The value $b_n = |n^P(q^n)|$ is the *breadth* chosen at that level. The tree root is therefore the planning node $n_0^P$ that receives the initial query $q$.

A research node $n^R(q_j^n)$ executes retrieval and localized reasoning for its particular subquery $q_j^n$:

$$n^R(q_j^n) \to (C_{q_j^n}, F_{q_j^n}), \tag{3}$$

producing local contexts $C_{q_j^n} \subseteq C$ and findings $F_{q_j^n} \subseteq F$. Optionally, a research node may trigger recursion by spawning a single child planning node that further decomposes $q_j^n$, after which the alternating pattern continues. The *depth* $d$ of the tree is defined by the number of research-node layers along the longest root-to-leaf path.

The final response $r_{\mathcal{T}}$ can then be synthesized as

$$r_{\mathcal{T}} = \sigma \left( q, \bigcup_{n_i \in N^R} C_i, \bigcup_{n_i \in N^R} F_i \right), \tag{4}$$

by aggregating each research node's local contexts and findings across the tree. The quality of the response can be subsequently measured by a utility function $U(r)$.

However, using fixed depths and breadths for the tree can lead to suboptimal performance: shallow trees might insufficiently explore the topic, while excessively deep or broad trees incur high costs with diminishing returns on quality. Therefore, the core challenge is to orchestrate the tree structure at runtime to maximize the response quality while adhering to a time budget $t_{\max}$. Formally, we aim to solve:

$$\max_{\mathcal{T}} U(r_{\mathcal{T}}) \quad \text{s.t.} \quad t(\mathcal{T}) \leq t_{\max}, \tag{5}$$

where $t(\mathcal{T})$ represents the total latency of the research process across all nodes and edges in the tree.

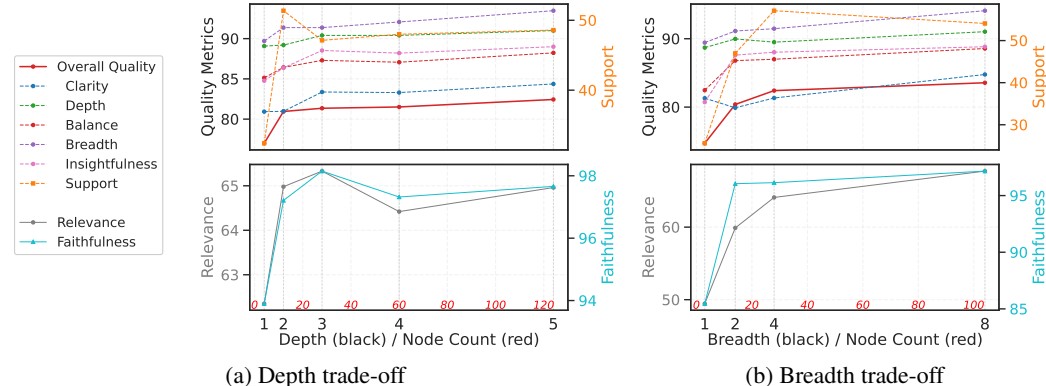

(a) Depth trade-off             (b) Breadth trade-off

Figure 2: **Trade-offs between deep research tree structure and response quality.** Left figure (a) varies *depth* (breadth fixed at 4) and right figure (b) varies *breadth* (depth fixed at 3); in each, the *top* plot shows *Quality* metrics with sub-metric *Support* on the right y-axis, while the *bottom* plot shows *Relevance* (left) and *Faithfulness* (right). The red labels along the x-axis give total node counts as a proxy for computational cost. Early increases raise quality, but gains saturate as cost escalates.

## 3.2 MOTIVATING EXPERIMENTS

Following the above formulation, we investigate how the tree structure can affect response quality in deep research tasks. We evaluated a tree-based deep research framework, GPT-Researcher, on a set of 100 complex queries randomly sampled from DeepResearchGym (Coelho et al., 2025). We varied the tree's maximum depth and breadth hyperparameters and measured performance across multiple metrics.

**Depth** In Figure 2(a), increasing depth from 1 to 2 yields the largest gain, with overall quality score rising sharply from 77.00 to 80.95. Beyond depth 3, the curves flatten. Extra depth produces only marginal quality improvements while node number grows exponentially. Notably, *Relevance* and *Faithfulness* peak at depth 3 and then decline, as deeper searches bring in peripheral sources and redundant materials, diluting core evidence and complicating the write-up compression.

**Breadth** A similar pattern can be observed when varying the breadth. In Figure 2(b), widening the tree from breadth 1 to 2 delivers a substantial quality gain with a moderate increase in nodes. Quality continues to improve up to breadth 4, after which the gains taper off.

To summarize, initial increases in depth or breadth are valuable, but returns diminish as node counts escalate. This highlights that a one-size-fits-all approach is inefficient. Adaptive planning is essential to tailor depth and breadth to each query's complexity, optimizing the quality-cost tradeoff.

## 4 FLASHRESEARCH

FlashResearch consists of three core components: (1) an **Adaptive Research Planner**, (2) a **Real-Time Orchestration Layer**, and (3) a **Multi-Dimensional Parallelization Framework**. Together, these components dynamically expand and prune the research tree $\mathcal{T}$ in real time and execute subtasks concurrently.

## 4.1 ADAPTIVE RESEARCH PLANNING

To efficiently navigate vast information spaces, it is essential to adapt the breadth and depth of research to the scope, nature, and complexity of each query. For example, broad queries like *"What is the impact of climate change?"* can be decomposed into multiple subqueries that address distinct aspects.In contrast, narrower questions like *"What's the process for developing film in a darkroom?"* require less exploration but demand greater focus and precision.

---

**Algorithm 1** Real-time Orchestration

---

1: **function** RESEARCHORCHESTRATOR($n_i^R$, $q^i$, $C_i$, $F_i$, $\Phi_{\min}$, $\Psi_{\min}$)
2:     $\tau_i$.should_terminate $\leftarrow$ False       ▷ Initialize termination flag for current subtree
3:     **Async** Execute research node $n_i^R$ , updating $C_i$, $F_i$ and parent node     ▷ Interruptible
4:     **Async** Plan child queries
5:     **for** each child query $q^j$ **do**
6:         **Async** Create $n_j^R$ with $C_j$, $F_j$         ▷ Speculative execution
7:         **Async** RESEARCHORCHESTRATOR($n_j^R$, $q^j$, $C_j$, $F_j$, $\Phi_{\min}$, $\Psi_{\min}$)   ▷ Recursive monitor
8:     **end for**
9:     **while** not $\tau_i$.should_terminate **do**       ▷ Continuous monitor at current level
10:         Update $C_i$, $F_i$ from node $n_i^R$ and descendant nodes $n_j^R$
11:         **Async** Evaluate $(\delta_i, \phi_i, \psi_i) \leftarrow \pi_o(q^i, C_i, F_i)$
12:         **if** $\delta_i = 0$ and $\phi_i \geq \Phi_{\min}$ and $\psi_i \geq \Psi_{\min}$ **then**
13:             $\tau_i$.should_terminate $\leftarrow$ True
14:             Interrupt node $n_i^R$ if ongoing         ▷ Early termination
15:             **for** each descendant $n_j^R$ **do**
16:                 Terminate $\tau_j$ recursively     ▷ Prune the descendant subtrees
17:             **end for**
18:         **end if**
19:         **if** $n_i^R$'s task completed and all children completed/terminated **then**
20:             $\tau_i$.should_terminate $\leftarrow$ True
21:         **end if**
22:     **end while**
23:     **return** Aggregated results from $C_i$, $F_i$ and children
24: **end function**

---

Therefore, we propose an adaptive research planner that decomposes the query $q^n$ into $b_n$ subqueries at each node $n^P \in N^P$, adjusting exploration breadth with a policy $\pi_b$ contextualized on the accumulated research findings $F$:

$$n^P(q^n) = \pi_b(q^n, F) = (b_n, \{q_1^n, \ldots, q_{b_n}^n\}), \tag{6}$$

where $b_n$ is the number of subqueries, $\{q_1^n, \ldots, q_{b_n}^n\}$ are the generated subqueries. A utility model guides the decision:

$$b_n = \underset{b \in [1, b_{\max}]}{\arg\max} \ \mathbb{E}[U(b \mid q^n, F)], \tag{7}$$

where $U$ represents a utility function estimating the expected information gain and relevance of decomposing into $b$ subqueries given the current context.

Once the breadth of exploration is set and the corresponding research is conducted, a policy $\pi_d$ assesses whether to deepen the current research path based on the localized research findings $F_i$ at each node $n_i^R \in N^R$:

$$\pi_d(q^i, F_i) = \mathbb{I}\{\mathbb{E}[U(F_{d+1} \mid q, F_i) - U(F_d)] > \tau\}, \tag{8}$$

where $F_d$ represents the accumulated research findings at depth $d$, $\tau$ is a threshold for diminishing returns, and the output is a binary decision whether further exploration yields sufficient information gains to justify the continued exploitation of the current research path. In our work, $\pi_b$ and $\pi_d$ are instantiated with LLM agents to support adaptive and intelligent decision-making (see Appendix A.1 for details), though in principle such policies could also be realized through supervised training or reinforcement learning.

### 4.2 REAL-TIME ORCHESTRATION

To address the dynamic and iterative nature of research, where priorities can shift as new evidence emerges, breadth planning conducted *prior-research* and depth planning performed *post-research* can be limiting. Furthermore, depth planning can also delay the exploitation of promising research paths until decisions are finalized.

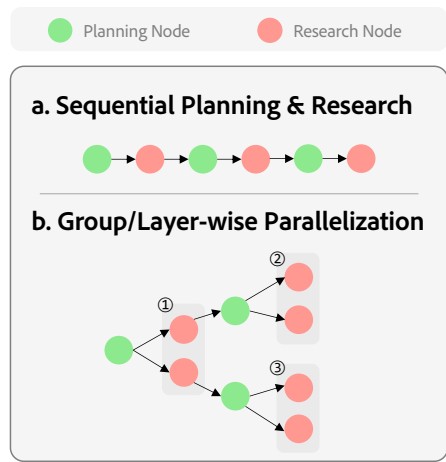
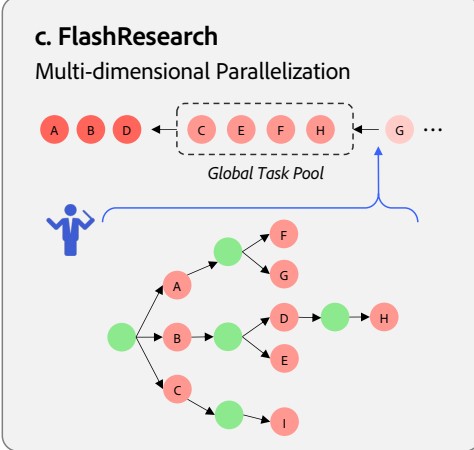

Figure 3: Sequential processing and group/layer parallelization introduce unnecessary latency by forcing nodes to wait for slow dependencies. FlashResearch supports multi-dimensional parallelization by submitting research nodes to a global task pool, where they are executed as soon as resources are available—so child nodes (e.g., D, E, F) can start immediately once their parents (A, B) finish, without being delayed by unrelated nodes like C.

To mitigate these, we introduce a real-time orchestration layer that dynamically manages the research tree $\mathcal{T}$ based on *mid-research* signals, enabling speculative execution and resource reallocation. Each research node $n^R(q^i)$ is continuously monitored by the orchestration policy $\pi_o$ based on the local query $q^i$, real-time context $C_i$, and accumulated findings $F_i$:

$$\pi_o(q_i, C_i, F_i) = (\delta_i, \phi_i, \psi_i) = \begin{cases} (0, \phi_i, \psi_i) & \text{if } \phi_i \geq \Phi_{\min} \text{ and } \psi_i \geq \Psi_{\min}, \\ (1, \phi_i, \psi_i) & \text{otherwise,} \end{cases} \quad (9)$$

where $\delta_i \in \{0, 1\}$ indicates task scheduling signals for continuation ($\delta_i = 1$) or termination ($\delta_i = 0$), based on whether the goal satisfaction score $\phi_i \in [0, 1]$ satisfies the goal satisfaction threshold $\Phi_{\min}$, and the quality score $\psi_i \in [0, 1]$ satisfies the quality threshold $\Psi_{\min}$. The policy is also implemented via an LLM agent, detailed in Appendix A.2.

More importantly, this mechanism enables *speculative execution*: child nodes can be spawned and deepen the tree without awaiting the parent's planning decision. Child nodes' findings update the parent's $C_i$ and $F_i$, even after the parent's research completes, enabling recursive and adaptive task management. As shown in Algorithm 1, upon evaluation at each hierarchy, low-yield nodes and their descendants are terminated early once the research goal is satisfied, pruning the subtree dynamically.

## 4.3 MULTI-DIMENSIONAL PARALLELIZATION

To maximize efficiency in traversing the adaptive research tree $\mathcal{T}$, FlashResearch incorporates a multi-dimensional parallel execution engine that enables concurrent processing across multiple axes: breadth (parallel subqueries at the same level), depth (speculative deepening of paths), and across the real-time orchestrators at different recursive hierarchies in Algorithm 1.

The engine operates by submitting all research nodes $n_i^R$ to a global asynchronous task pool as soon as they are planned and orchestrated. Each node is parameterized by its local query $q^i$, depth $d_i$, parent identifier $p_i$, and a unique task identifier $t_i$. Dependencies are enforced dynamically: a child node $n_j^R$ (with $p_j = t_i$) becomes eligible for execution only once its parent $n_i^R$ completes its initial research phase, but speculative spawning allows planning and partial execution to begin earlier under the real-time orchestrator's guidance.

FlashResearch's engine leverages non-blocking asynchronous calls, allowing tasks to progress independently. This approach mitigates bottlenecks inherent in sequential or coarse-grained parallel execution, where nodes must wait for unrelated dependencies to complete. For instance, as illustrated in Figure 3, child nodes (e.g., D, E, F) can initiate immediately upon their respective parents' (A, B) completion, without delays from slower siblings like C.

# 5 EXPERIMENTS

## 5.1 BENCHMARKS AND EVALUATION METRICS

We evaluate our method on two recent deep research benchmarks.

**DeepResearchGym** (Coelho et al., 2025) is an open-source evaluation sandbox for deep research systems. The benchmark consists of the top 1,000 complex, high-engagement non-factoid queries from the Researchy Questions dataset (Rosset et al., 2025). We randomly sampled 100 for evaluation. It employs an LLM-as-a-judge protocol to assess generated reports along three dimensions:

- **Quality** Rates organization, clarity, and coherence of the synthesis. Prefers well-structured, readable reports that integrate evidence into concise, actionable takeaways.
- **Relevance** Judges whether the report directly answers the user's intent, covering the key sub-questions and constraints in the prompt. Penalizes omissions and off-topic content.
- **Faithfulness** Assesses whether the cited evidence supports claims. Rewards correctly grounded statements and flags contradictions, unsupported claims, or hallucinations.

**DeepResearch Bench** (Du et al., 2025) comprises 100 PhD-level research tasks across 22 distinct fields. These tasks were designed by domain experts based on a statistical analysis of over 96,000 real-world user queries from web search–enabled LLM interactions. The benchmark includes 50 English and 50 Chinese tasks. We focus on the English subset in evaluation. The benchmark proposes two evaluation frameworks: RACE for report quality and FACT for citation trustworthiness.

- **RACE**:
  - **Comprehensiveness**: Evaluates thorough coverage of relevant aspects, including diverse perspectives and key subtopics, ensuring understanding without omissions.
  - **Depth**: Assesses level of detail, analysis, and insights beyond surface-level, including causes, impacts, and trends.
  - **Instruction following**: Checks adherence to query requirements, ensuring alignment with intent by following the topic and answering directly.
  - **Readability**: Assesses clarity through structure, language, and ease of understanding.
- **FACT**:
  - **Effective citation count**: Measures factual abundance by counting the number of unique, relevant citations that effectively support key statements in the report.
  - **Citation accuracy**: Evaluates citation trustworthiness by assessing the proportion of citations that effectively support the referenced claims.

## 5.2 EVALUATION SETUP

To ensure a fair comparison, we build FlashResearch on top of GPT-Researcher's agentic workflow (Elovic, 2023), and evaluate FlashResearch against the original GPT-Researcher, using FineWeb (Penedo et al., 2024) as the static web corpora to improve reproducibility. On DeepResearchGym, we fix maximum execution times at 2 and 10 minutes to reflect realistic usage scenarios:

- The 2-minute cutoff reflects human multitasking behavior, where information workers spend an average of ∼2–3 minutes on events or tools before task switching (González & Mark, 2004).
- The 10-minute threshold matches the average duration of a "working sphere" (González & Mark, 2004) and is supported by evidence from high-performance computing (Schlagkamp & Renker, 2015) and crowdsourcing tasks (Bernstein et al., 2011), indicating that a 10-minute window preserves task continuity without losing human-in-the-loop coordination.

## 5.3 RESULTS

**DeepResearchGym.** We evaluated FlashResearch against both GPT-Researcher and an ablated variant, FlashResearch*, which omits adaptive research planning and real-time orchestration. The results in Table 1 show that FlashResearch consistently delivers superior throughput, processing substantially more research than the GPT-Researcher baseline (up to $4.11\times$ in the 10-minute setup).

Table 1: Evaluation of deep research frameworks on DeepResearchGym under fixed time budgets. Scores are assessed by `gpt-4.1-mini-2025-04-14` and averaged over 5 runs. All metrics reported with 95% confidence intervals. FlashResearch (-AP) refers to the ablation without adaptive planning. FlashResearch (-AP, -RO) refers to the ablation without adaptive planning and real-time orchestration, previously marked as FlashResearch*.

| | Throughput | Quality | | | | | | | Relevance | | Faithfulness |
|---|---|---|---|---|---|---|---|---|---|---|---|
| | # Nodes | Overall | Clarity | Depth | Balance | Breadth | Support | Insight | KPR | KPC (↓) | Cit. Recall |
| **2 minutes** | | | | | | | | | | | |
| GPT-Researcher | 8.00 ± 0.18 | 74.06 ± 0.88 | 78.66 ± 1.94 | 88.88 ± 0.70 | 84.70 ± 0.80 | 90.64 ± 0.27 | 21.22 ± 3.39 | 80.26 ± 1.88 | 54.50 ± 2.16 | 0.76 ± 0.35 | 85.54 ± 1.77 |
| FlashResearch | 19.42 ± 1.31 | **79.78 ± 0.80** | **83.88 ± 1.06** | 90.26 ± 0.22 | **87.36 ± 0.47** | 91.10 ± 0.27 | **40.10 ± 3.91** | 85.98 ± 1.01 | 62.29 ± 2.00 | 0.73 ± 0.26 | **94.71 ± 0.87** |
| FlashResearch (-AP) | 8.94 ± 0.97 | 79.44 ± 0.76 | 86.64 ± 0.68 | **90.58 ± 0.24** | 87.26 ± 0.72 | **92.22 ± 0.39** | 33.80 ± 3.88 | 86.16 ± 1.15 | 62.73 ± 1.99 | 0.77 ± 0.25 | 84.88 ± 1.78 |
| FlashResearch (-AP, -RO) | **21.14 ± 1.01** | 78.88 ± 0.96 | 82.10 ± 1.37 | 89.04 ± 0.75 | 85.36 ± 1.02 | 89.94 ± 0.92 | 39.62 ± 3.91 | **87.22 ± 0.86** | **66.72 ± 2.01** | **0.43 ± 0.21** | 94.36 ± 0.86 |
| **10 minutes** | | | | | | | | | | | |
| GPT-Researcher | 23.94 ± 0.83 | 79.52 ± 0.80 | 82.78 ± 1.27 | 89.96 ± 0.41 | 86.60 ± 0.51 | 90.64 ± 0.44 | 39.48 ± 3.91 | 87.64 ± 0.78 | 63.88 ± 1.88 | 0.61 ± 0.27 | 95.53 ± 0.80 |
| FlashResearch | **98.43 ± 0.19** | **83.90 ± 0.78** | 83.08 ± 1.25 | 90.52 ± 0.43 | **88.08 ± 0.55** | **94.42 ± 0.44** | **58.10 ± 3.81** | 89.18 ± 0.46 | 66.16 ± 1.93 | 0.55 ± 0.22 | 95.96 ± 1.03 |
| FlashResearch (-AP) | 62.32 ± 9.84 | 83.12 ± 0.79 | 82.80 ± 1.30 | **90.60 ± 0.31** | 87.86 ± 0.59 | 93.88 ± 0.43 | 54.20 ± 3.94 | **89.38 ± 0.25** | **68.42 ± 1.91** | 0.56 ± 0.23 | **97.57 ± 0.49** |
| FlashResearch (-AP, -RO) | 68.00 ± 0.00 | 82.69 ± 1.04 | **83.28 ± 1.32** | 89.42 ± 0.95 | 87.44 ± 0.87 | 92.46 ± 0.76 | 56.24 ± 3.85 | 87.32 ± 1.14 | 66.44 ± 1.98 | 0.51 ± 0.21 | 96.52 ± 0.65 |

Table 2: Overall evaluation results on DeepResearch Bench under flexible time budgets, judged by `Gemini-2.5-flash` and `Gemini-2.5-pro`. Commercial deep research agents' results are directly from the DeepResearch Bench Leaderboard[1], which has no latency statistics reported.

| Method | Throughput | | RACE | | | | | FACT | |
|---|---|---|---|---|---|---|---|---|---|
| | # Nodes | Latency | Overall | Comp. | Depth | Inst. | Read. | Cit. Acc. | Eff. Cit. |
| Grok Deeper Search | - | - | 38.22 | 36.08 | 30.89 | 46.59 | 42.17 | 73.08 | 8.58 |
| Perplexity Research | - | - | 40.46 | 39.10 | 35.65 | 46.11 | 43.08 | 82.63 | 31.20 |
| OpenAI Deep Research | - | - | 46.45 | 46.46 | 43.73 | 49.39 | 47.22 | 75.01 | 39.79 |
| Gemini-2.5-Pro Deep Research | - | - | 49.71 | 49.51 | 49.45 | 50.12 | 50.00 | 78.30 | 165.34 |
| GPT-Researcher | 23.12 | 554.41 s | 41.15 | 38.58 | 37.55 | 46.03 | 45.62 | 65.58 | 9.40 |
| FlashResearch* | 27.88 | **207.06** s | 41.33 | 38.61 | 38.09 | 46.01 | 45.80 | 70.06 | 17.35 |
| FlashResearch | **39.30** | 367.88 s | **41.92** | **39.55** | **38.61** | **46.36** | **45.83** | 58.25 | 22.94 |

This efficiency gain arises from adaptive planning and real-time orchestration, which enable speculative execution without compromising quality.

Beyond throughput, FlashResearch also improves overall response quality, with clear improvements in balance, breadth, and insight metrics. These gains highlight its ability to maintain a robust trade-off between expansive coverage and focused analysis, particularly under tight time constraints. Notably, the overall quality of FlashResearch with 2-minute execution even surpasses that of the GPT-Researcher baseline with 10 minutes, demonstrating a 5× speed-up while preserving quality. Overall, these results underscore FlashResearch 's capacity for dynamic adaptation, producing more comprehensive and higher-quality research outputs under constrained budgets.

**DeepResearch Bench.** To better assess our system's performance relative to other deep research agents, we also evaluate it under flexible time budgets on DeepResearch Bench. As shown in Table 2, FlashResearch achieves substantial efficiency gains, processing 39.3 nodes on average while reducing latency by 1.51× compared to the GPT-Researcher baseline.

In terms of quality, our proposed framework also consistently improves across all RACE sub-metrics. Compared to the ablated method, FlashResearch incurs higher latency but conducts more research at a comparable throughput, highlighting a favorable trade-off between efficiency and comprehensiveness. Importantly, FlashResearch also achieves a performance competitive with commercial systems like Grok Deeper Search and Perplexity Research, narrowing the gap with state-of-the-art proprietary agents. Beyond quantitative metrics, we also provide a detailed case analysis of FlashResearch's adaptability across diverse query conditions in Appendix B.

---

[1] `https://huggingface.co/spaces/Ayanami0730/DeepResearch-Leaderboard`

# 6 CONCLUSION

We present FlashResearch to enhance the efficiency of deep research tasks through adaptive planning, real-time orchestration, and multi-dimensional parallelization. By formulating deep research as a tree-structured process and dynamically allocating resources to promising paths, FlashResearch achieves significant improvements in research throughput and response quality, as demonstrated through extensive evaluations on two deep research benchmarks. Future work includes incorporating richer modalities beyond text and exploring tighter integration with human-in-the-loop monitoring and interruption to further improve transparency and usability.

## REPRODUCIBILITY STATEMENT

Additional implementation details and experimental setups are included in the Appendix A. The complete source code and instructions for reproducing all experiments are available at the following anonymous repository: `https://anonymous.4open.science/r/FlashResearch/`.

## ETHICS STATEMENT

This work does not involve human subjects, private data, or sensitive information. Experiments were conducted using publicly available datasets. While our framework aims to improve the efficiency and quality of deep research systems, we acknowledge the broader risks of misuse, including the potential amplification of biased or unreliable information. Responsible deployment requires careful selection of data sources, robust fact-checking, and adherence to ethical standards.

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

## A    IMPLEMENTATION DETAILS

### A.1    ADAPTIVE RESEARCH PLANNING

The adaptive research planner dynamically decomposes queries into subqueries using LLM-based policies $\pi_b$ and $\pi_d$. The following prompt is used by $\pi_b$ to determine the optimal number of subqueries $b_n$, balancing exploration breadth based on the utility model in Equation (7):

---

**Prompt 1**

You are an expert researcher generating search queries. Your task is to determine the OPTIMAL number of clear, non-overlapping search queries.
EFFICIENCY IS CRITICAL: More subqueries do not necessarily lead to better research. Minimize waste and redundancy. Highly specific queries need fewer subqueries. Broad topics may need more.

SUBQUERY REQUIREMENTS:
- Do not exceed [max_breadth+flex_breadth] subqueries
- Keep queries clear and concise
- Make each subquery target a DISTINCT aspect
- Avoid near-duplicates and trivial variants
- Prefer fewer subqueries if coverage is maintained
- Ensure queries are relevant to the high-level research goal: [initial_query]
- Exclude overlap with existing learnings: [accumulated_learnings]

---

This prompt enables the generation of subqueries, adjusting the breadth $b_n$ based on the query and accumulated research findings $F$, as described in Equation (6).

### A.2    REAL-TIME ORCHESTRATION

The real-time orchestration layer monitors task execution with a policy $\pi_o$ to assess goal satisfaction and quality. The following prompt implements $\pi_o$ to compute goal satisfaction ($\phi_n$) and quality ($\psi_n$) scores, enabling speculative execution and early termination as per Equation (9):

---

**Prompt 2**

You are an expert research quality evaluator. Determine if a research goal has been sufficiently satisfied based on current findings.

EVALUATION CRITERIA:
1. GOAL COVERAGE: Does the research adequately address the stated goal?
2. INFORMATION QUALITY: Are the findings comprehensive and reliable?
3. DEPTH SUFFICIENCY: Is there enough detail to answer the research question?
4. SOURCE DIVERSITY: Are findings from multiple credible sources?
5. COMPLETENESS: Are major aspects of the topic covered?

SATISFACTION SCORE:
- HIGH SATISFACTION (0.8-1.0): Goal fully satisfied, comprehensive coverage
- MEDIUM SATISFACTION (0.5-0.8): Goal mostly satisfied, minor gaps acceptable
- LOW SATISFACTION (0.3-0.5): Goal partially satisfied, significant gaps remain
- INSUFFICIENT (0.0-0.3): Goal not satisfied, major research needed

QUALITY SCORING:
- EXCELLENT (0.8-1.0): Comprehensive, well-sourced, detailed
- GOOD (0.5-0.8): Adequate coverage, some depth
- FAIR (0.3-0.5): Basic coverage, limited depth
- POOR (0.0-0.3): Insufficient information

Be conservative - only mark as satisfied if the research truly addresses the goal comprehensively.

---

Table 3: Statistical comparison between the sampled set's 100 questions and the whole DeepResearchGym's 1000 questions, based on the characterizations data from the original Researchy Questions dataset (Rosset et al., 2025). MW represents the Mann-Whitney U test. KS represents the Kolmogorov-Smirnov test. TOST stands for the Equivalence Testing with Two One-Sided Tests.

| Score Type | Sampled Set
Mean ± SD | DeepResearchGym
Mean ± SD | MW
$p$-value | KS
$p$-value | Cohen's d | TOST $p$-value
($\delta$=0.5×SD) | Equiv? |
|---|---|---|---|---|---|---|---|
| Decompositional | 0.735 ± 0.088 | 0.734 ± 0.087 | 0.9563 | 0.9348 | 0.013 | 0.0000 | Yes |
| Nonfactoid | 1.022 ± 0.085 | 1.017 ± 0.083 | 0.2800 | 0.5207 | 0.070 | 0.0000 | Yes |
| Ambiguous | 0.710 ± 0.697 | 0.797 ± 0.735 | 0.2819 | 0.9898 | -0.119 | 0.0001 | Yes |
| Incompleteness | 1.390 ± 1.580 | 1.527 ± 1.546 | 0.2330 | 0.9433 | -0.088 | 0.0000 | Yes |
| Assumptive | 0.920 ± 2.226 | 0.964 ± 2.122 | 0.1772 | 0.6794 | -0.021 | 0.0000 | Yes |
| Multi-faceted | 7.140 ± 1.149 | 7.115 ± 1.250 | 0.9458 | 1.0000 | 0.020 | 0.0000 | Yes |
| Knowledge-intensive | 6.510 ± 1.559 | 6.822 ± 1.307 | 0.0526 | 0.5990 | -0.234 | 0.0057 | Yes |
| Subjective | 5.060 ± 2.716 | 4.999 ± 2.666 | 0.6871 | 0.9433 | 0.023 | 0.0000 | Yes |
| Reasoning-intensive | 6.480 ± 1.144 | 6.600 ± 1.188 | 0.1535 | 0.3537 | -0.101 | 0.0001 | Yes |
| Harmful | 0.000 ± 0.000 | 0.000 ± 0.000 | 1.0000 | 1.0000 | — | — | — |

Note: TOST $p$-value $< 0.05$ indicates two groups are statistically equivalent within margin $\delta$.

Table 4: Inter-rater reliability analysis using Intraclass Correlation Coefficient (ICC) across three systems under two time constraints. Higher ICC values indicate stronger agreement between multiple runs of the LLM judge.

| Time Budget | System | Quality | KPR | KPC | Citation Recall |
|---|---|---|---|---|---|
| 2 minutes | GPT-Researcher | 0.917** | 0.995** | 0.968** | 0.979** |
| | FlashResearch* | 0.909** | 0.996** | 0.991** | 0.932** |
| | FlashResearch | 0.890* | 0.994** | 0.966** | 0.961** |
| 10 minutes | GPT-Researcher | 0.859* | 0.995** | 0.994** | 0.985** |
| | FlashResearch* | 0.942** | 0.996** | 0.975** | 0.969** |
| | FlashResearch | 0.882* | 0.991** | 0.962** | 0.979** |

Note: * Good reliability ($0.75 \leq$ ICC $< 0.90$), ** Excellent reliability (ICC $\geq 0.90$).

This prompt facilitates the evaluation of task outputs against thresholds $\Phi_{min}$ and $\Psi_{min}$, pruning low-yield paths dynamically. In our experiments, both thresholds $\Phi_{min}$ and $\Psi_{min}$ are set to 0.8.

## A.3 EXPERIMENTAL SETUPS

For model configuration, we use `gpt-4.1-mini-2025-04-14` for the main research processing, and `o3-mini-2025-01-31` for major policy decisions, including adaptive research planning and real-time orchestration.

To ensure fair comparisons among deep research systems, we impose a maximum execution time for research trees. Once the time cut-off is reached, the research process terminates immediately, and the system generates a response based on the findings and context gathered up to that point. To allow all systems to fully utilize their time budgets, we set the maximum tree depth to 10 and the maximum breadth to 4 within the GPT-Researcher framework. To provide additional flexibility, the adaptive research planning module may expand the breadth up to 6 when necessary.

Additionally, to control the computational costs introduced by the real-time orchestration layer, we set an interval of 8 seconds between successive evaluations of goal satisfaction and research quality.

## B CASE ANALYSIS

To illustrate how FlashResearch adapts its research process to different query conditions, we present the research trees from three cases in DeepResearch Bench. For a controlled comparison, we standardize the time cutoff to 2 minutes across all cases.

Table 5: Statistical significance of performance metric changes from 2-minute to 10-minute time budgets, evaluated via independent samples t-tests ($\Delta$ represents mean change).

| Metric | GPT-Researcher | | | FlashResearch* | | | FlashResearch | | |
|---|---|---|---|---|---|---|---|---|---|
| | $\Delta$ | $t$-value | $p$-value | $\Delta$ | $t$-value | $p$-value | $\Delta$ | $t$-value | $p$-value |
| **Quality** | | | | | | | | | |
| Overall | +5.46*** | 9.03 | <0.001 | +3.81*** | 5.27 | <0.001 | +4.12*** | 7.21 | <0.001 |
| Clarity | +4.12*** | 3.49 | <0.001 | +1.18 | 1.22 | 0.224 | -0.80 | -0.96 | 0.339 |
| Depth | +1.08** | 2.60 | 0.010 | +0.38 | 0.62 | 0.538 | +0.26 | 1.06 | 0.288 |
| Balance | +1.90*** | 3.93 | <0.001 | +2.08** | 3.04 | 0.002 | +0.72 | 1.95 | 0.052 |
| Breadth | +0.00 | 0.00 | 1.000 | +2.52*** | 4.15 | <0.001 | +3.32*** | 12.64 | <0.001 |
| Support | +18.26*** | 6.90 | <0.001 | +16.62*** | 5.93 | <0.001 | +18.00*** | 6.45 | <0.001 |
| Insightfulness | +7.38*** | 7.10 | <0.001 | +0.10 | 0.14 | 0.891 | +3.20*** | 5.67 | <0.001 |
| **Relevance** | | | | | | | | | |
| KPR | +9.37*** | 6.40 | <0.001 | -0.28 | -0.20 | 0.845 | +3.88** | 2.73 | 0.006 |
| KPC ($\downarrow$) | -0.15 | -0.66 | 0.508 | +0.08 | 0.52 | 0.604 | -0.18 | -1.06 | 0.289 |
| **Faithfulness** | | | | | | | | | |
| Citation Recall | +9.99*** | 10.09 | <0.001 | +2.16*** | 3.92 | <0.001 | +1.25 | 1.81 | 0.070 |

Note: Significance levels: *** $p < 0.001$, ** $p < 0.01$, * $p < 0.05$.

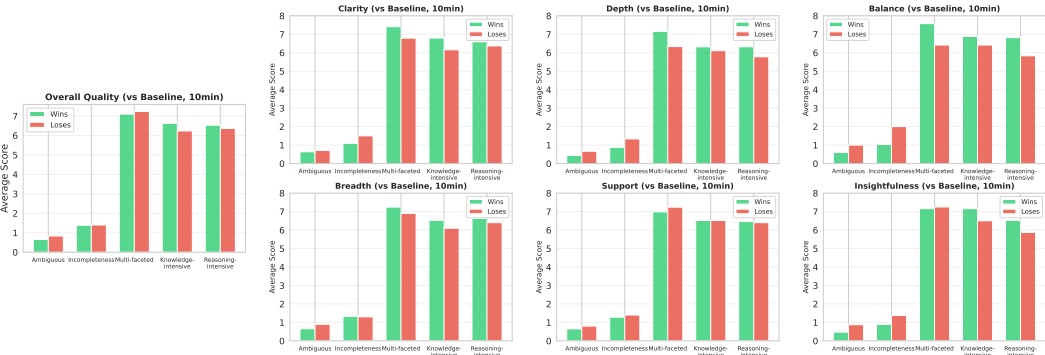

Figure 4: FlashResearch performance analysis (v.s. GPT-Researcher baseline, 10 min). We compare the average characterization scores for the 70 questions where FlashResearch *Wins* (outperforms the baseline) against the 30 questions where it *Loses*. Scores are plotted across five characterization dimensions (x-axis) from the Researchy Questions dataset (Rosset et al., 2025) and evaluated with respect to six quality sub-metrics.

In Case 1 (Figure 5), the query concerns a broad topic: *investigating current non-alcoholic cocktails*. The tree expands widely across multiple layers of branching, encompassing diverse angles such as ingredient sourcing, sustainability, AI-driven methods, and integration with fine dining. This highlights FlashResearch 's ability to surface complementary perspectives when the query is exploratory and open-ended.

In contrast, Case 2 yields a more compact tree in Figure 6 for a narrow, domain-specific query on *cislunar situational awareness*. The decomposition focuses on specialized aspects, including hybrid sensor fusion, bias-correction methods, and federated filtering. With the research goal achievable through limited exploration, FlashResearch terminates at depth 2 to avoid redundant effort.

Finally, Case 3 depicts a common scenario where users explicitly specify particular focuses for a deeper investigation. As shown in Figure 7, FlashResearch tailors the tree to these requirements, deepening its analysis of *AI's disruptions across various industries*. In particular, we examined the execution logs and analyzed the reasoning traces to gain a clearer understanding of the decision-making process during real-time orchestration. The user query explicitly requests "*only cites high-quality, English-language journal articles*". Consequently, when the orchestrator detects that a node does not satisfy this research goal, it terminates the branch early to avoid unnecessary computation. In this specific case, the orchestrator's reasoning trace is shown below.

Table 6: Evaluation of deep research frameworks on DeepResearchGym with 10-minute time budget using the open-source `Qwen3-235B-A22B-Instruct-2507` model for all the operations. Scores are assessed by `gpt-4.1-mini-2025-04-14` and averaged over 5 runs. All metrics reported with 95% confidence intervals.

| | Throughput | Quality | | | | | | | Relevance | | Faithfulness |
|---|---|---|---|---|---|---|---|---|---|---|---|
| | # Nodes | Overall | Clarity | Depth | Balance | Breadth | Support | Insight | KPR | KPC (↓) | Cit. Recall |
| **2 minutes** | | | | | | | | | | | |
| Baseline | 3.94 ± 0.18 | 82.26 ± 0.92 | 87.68 ± 1.26 | 91.38 ± 0.39 | 87.54 ± 0.93 | 93.94 ± 0.69 | 46.80 ± 4.20 | 86.20 ± 1.25 | 65.25 ± 2.09 | **0.47 ± 0.21** | 79.83 ± 2.02 |
| FlashResearch | 10.90 ± 0.69 | 84.91 ± 0.76 | 89.22 ± 0.62 | 96.72 ± 0.41 | 89.00 ± 0.66 | 96.22 ± 0.43 | 48.16 ± 4.04 | 90.12 ± 0.43 | 69.37 ± 2.00 | 0.93 ± 0.29 | 74.56 ± 1.65 |
| **10 minutes** | | | | | | | | | | | |
| Baseline | 15.14 ± 0.31 | 83.48 ± 0.79 | **85.56 ± 0.97** | 90.14 ± 0.31 | 87.58 ± 0.67 | 91.04 ± 0.50 | 58.92 ± 3.82 | 87.66 ± 0.98 | 68.59 ± 1.88 | **0.47 ± 0.26** | 95.85 ± 0.86 |
| FlashResearch | **68.30 ± 10.04** | **87.68 ± 0.60** | 83.04 ± 0.99 | **93.84 ± 0.43** | 88.38 ± 0.57 | 94.44 ± 0.44 | 76.64 ± 3.04 | 89.76 ± 0.29 | 69.99 ± 2.05 | 0.98 ± 0.31 | 93.78 ± 0.82 |

Table 7: LLM token usage and API cost analysis on DeepResearchGym with different time budgets. All values are averaged over 100 questions. Orchestration metrics only apply to FlashResearch.

| | Orchestration Usage | | | Total Usage | | |
|---|---|---|---|---|---|---|
| | Input Tokens | Output Tokens | Cost ($) | Input Tokens | Output Tokens | Total Cost ($) |
| **2 minutes** | | | | | | |
| GPT-Researcher | – | – | – | 97,170 | 5,428 | 0.1258 |
| FlashResearch* | – | – | – | 176,147 | 9,355 | 0.2218 |
| FlashResearch | 9,632 | 1,788 | 0.0185 | 155,409 | 10,031 | 0.2071 |
| **10 minutes** | | | | | | |
| GPT-Researcher | – | – | – | 299,093 | 18,432 | 0.3941 |
| FlashResearch* | – | – | – | 424,488 | 28,496 | 0.5629 |
| FlashResearch | 25,646 | 4,756 | 0.0491 | 487,023 | 36,409 | 0.6811 |

---

**Reasoning Trace 1**

The current research progress mainly comprises news articles, reports, and policy discussions rather than high-quality, peer-reviewed English journal articles specifically focused on AI-driven labor market restructuring. While the materials are relevant to the broader topic, they do not meet the stringent criteria of academic journal quality necessary to fully satisfy the research goal. The findings provide useful context and cover significant aspects of AI's impact on the labor market, but they lack the depth, methodological rigor, and academic sourcing expected from high-quality journal articles. Consequently, although there is moderate progress on the subject, key elements of the goal remain unmet, suggesting that it would be more cost-efficient to redirect or deepen the research efforts rather than continuing on the current path.

---

Collectively, these cases demonstrate FlashResearch 's adaptability: expanding broadly for open domains, conserving resources when goals can be met with focused research, and tailoring scope when users impose explicit constraints. More importantly, since thinking models are used for the planning and orchestration, the framework generates explicit reasoning traces that facilitate the interpretation of decisions and enhance transparency in error tracking.

## C  USE OF LLMS

We used LLMs as assistive tools for (i) polishing the manuscript; (ii) revising the phrasing of aforementioned LLM prompts, and (iii) refining code. LLMs did not contribute to research ideation, experimentation, or the formulation of claims.

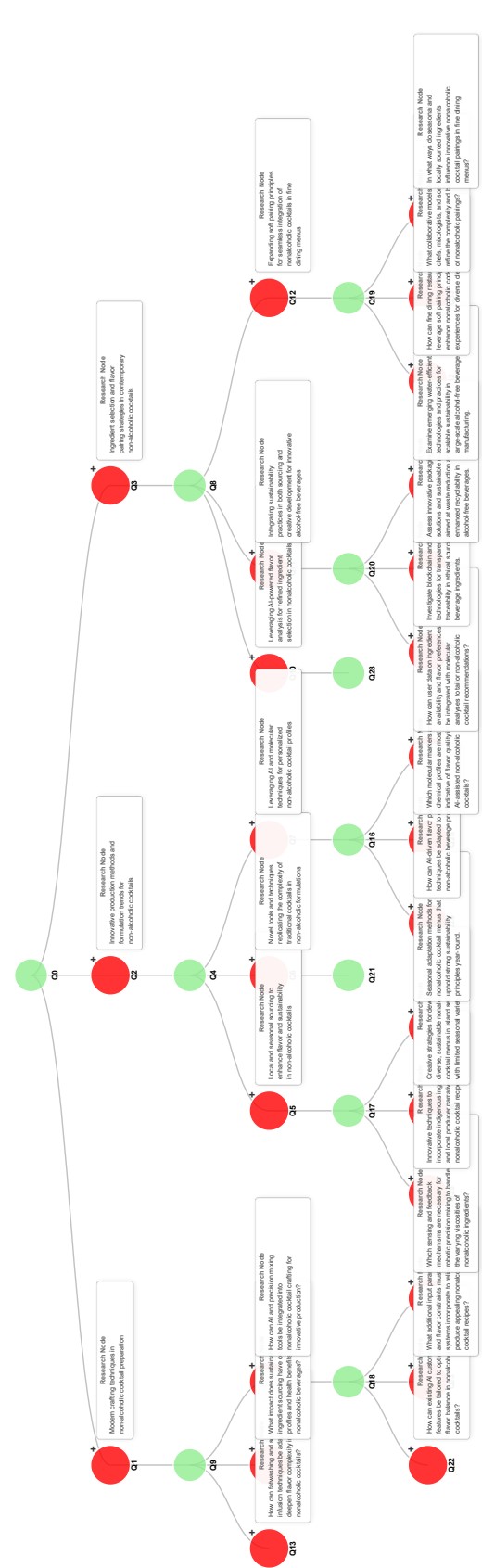

Figure 5: Research tree generated by FlashResearch for a broad-topic query: *"Research Topic:: Crafting Techniques for Non-Alcoholic Cocktails. Objective: Investigate current non-alcoholic cocktails to discover innovative production methods and formulations."*

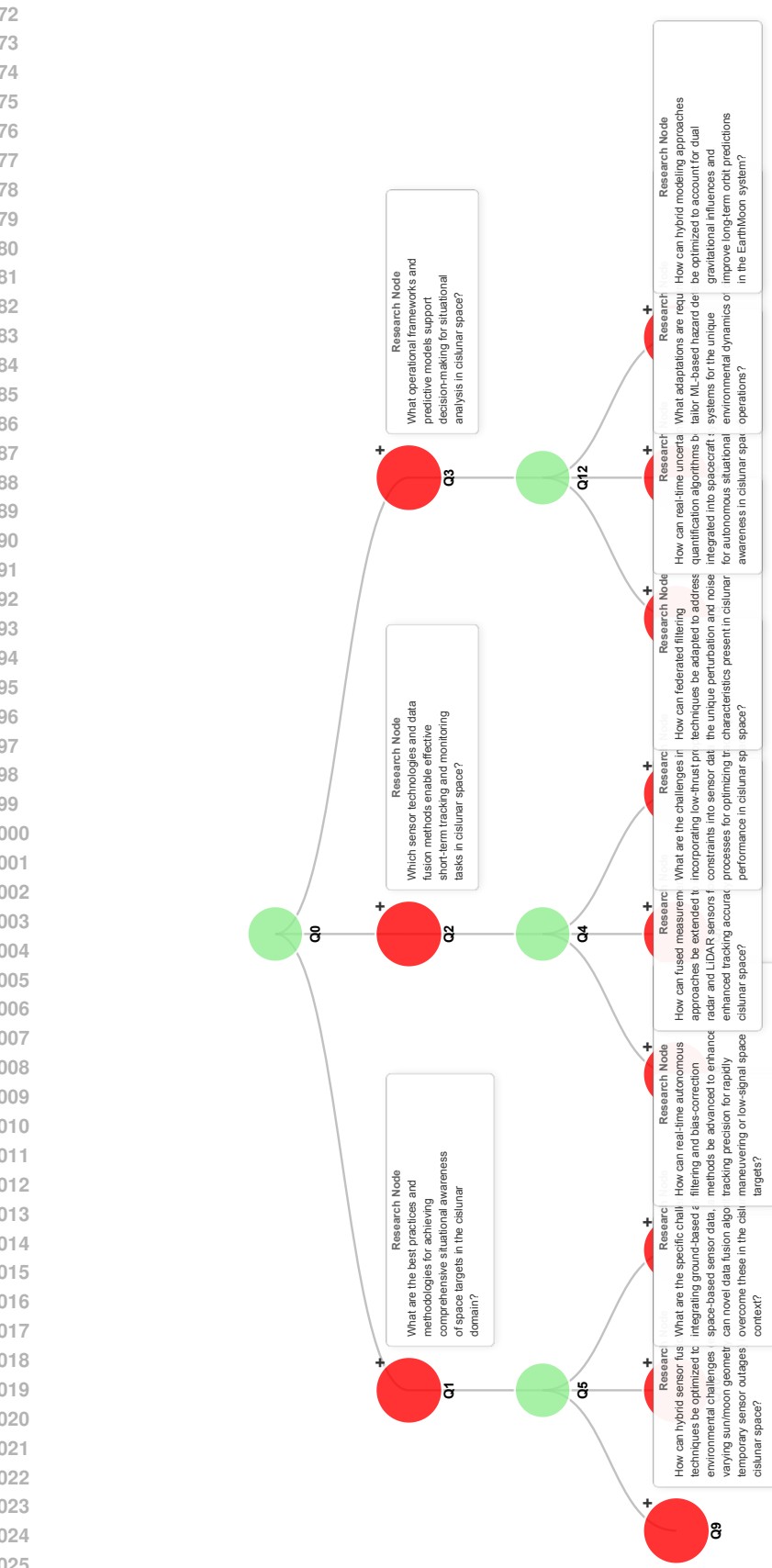

Figure 6: Research tree generated by FlashResearch for a narrow, domain-specific query: *"How to conduct comprehensive and accurate situational awareness of space targets in the cislunar space, and support the effectiveness of short-term cislunar space tracking and monitoring tasks?"*

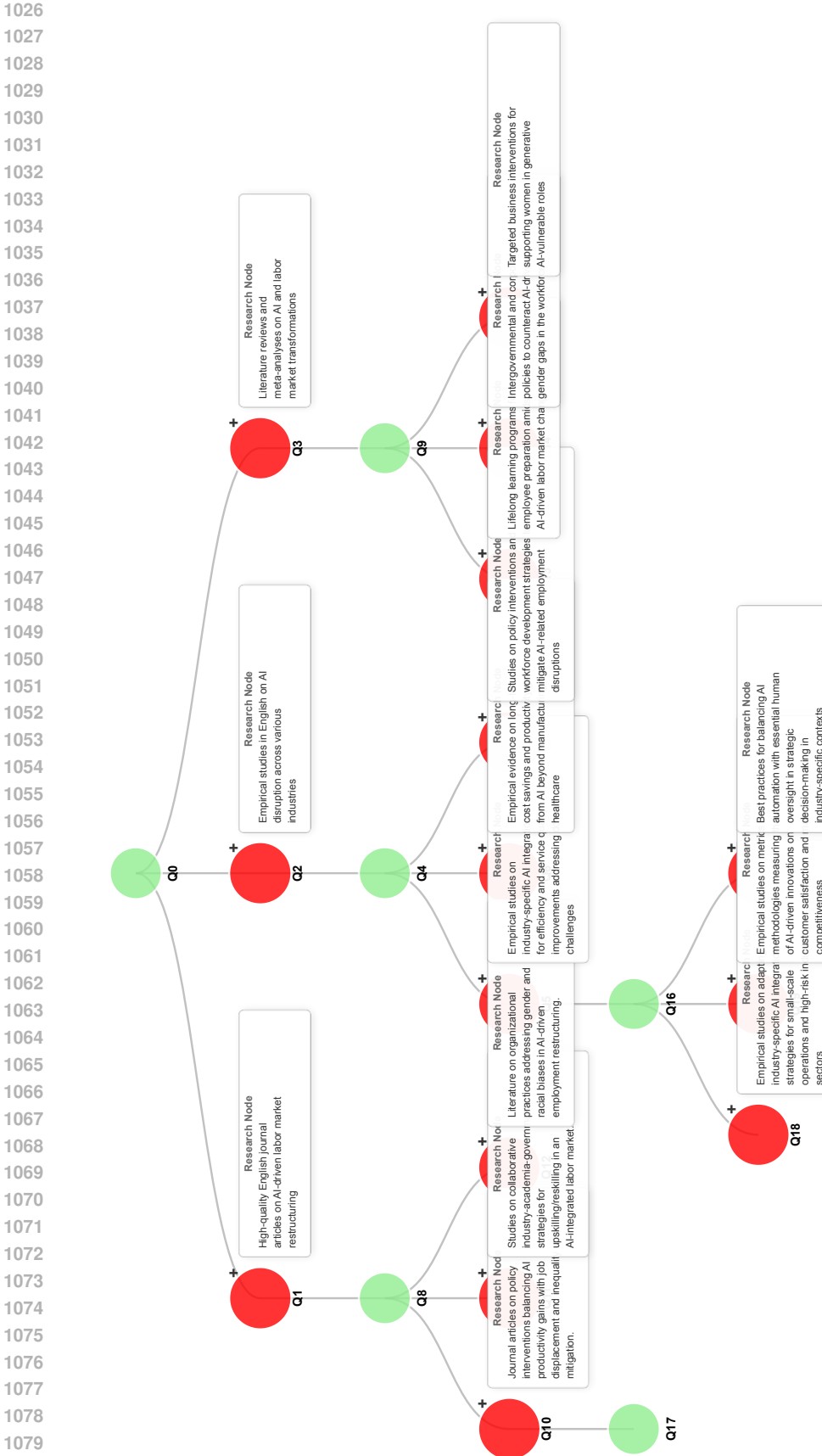

Figure 7: Research tree generated by FlashResearch for a user-focused query with explicit demands: *"Please write a literature review on the restructuring impact of Artificial Intelligence (AI) on the labor market. Focus on how AI, as a key driver of the Fourth Industrial Revolution, is causing significant disruptions and affecting various industries. Ensure the review only cites high-quality, English-language journal articles."*

