# OpenReview forum: "FlashResearch: Real-time Agent Orchestration for Efficient Deep Research"
_ICLR.cc/2026/Conference — Submitted to ICLR 2026_

### Official Review · Reviewer_YF1V · 2025-10-22

**Soundness:** 3
**Presentation:** 2
**Contribution:** 2
**Rating:** 6
**Confidence:** 3

**Summary:**

FlashResearch converts deep research into a tree-structured, real-time orchestrated search process, parallelizes sub-queries and prunes low-value branches under a hard time budget. An adaptive planner chooses breadth and depth per node, while an asynchronous execution engine enables speculative deepening and early termination. Evaluated on DeepResearchGym and DeepResearch Bench (though a little bit narrow), the system processes up to 4× more nodes than the GPT-Researcher baseline and yields slightly higher report scores within the same wall-clock limit.

**Strengths:**

- Demonstrates measurable throughput-quality lift over a strong retrieval baseline under fixed time budgets.
- Provides a clean, asynchronous tree abstraction that cleanly separates planning, orchestration, and execution concerns.

**Weaknesses:**

1. FlashResearch appears to focus primarily on an offline setting, where data are collected from a local corpus (for instance, the setting in Figure 1 seems to be strongly related to DeepResearchGym). However, I understand that one of the major challenges of deep research actually lies in tool selection and allocation within online web search scenarios—different task types may require different tools (e.g., some can be efficiently solved via a Wikipedia Search API, while others may benefit from Playwright-based visual browsing). Yet, I did not observe any adaptation or discussion regarding this aspect. I understand that DeepResearchGym may only require a single retriever tool, but what kinds of tools does DeepResearchBench actually provide?
2. The paper’s survey of current state-of-the-art deep research methods is somehow insufficient. The cited works (e.g., AFlow, Flow, and EvoFlow) remain confined to traditional math or coding tasks, whereas there now exist many multi-agent approaches specifically designed for deep research, including but not limited to [1, 2, 3, 4]. I believe it is necessary to mention, cite, and even compare these works.
3. In Line 135, the authors claim that most existing agentic workflows “lack support for real-time replanning.” This statement is inaccurate, as a number of studies have already demonstrated the ability to dynamically adjust plans in real time [5, 6]. The authors should take this into account; there is also a growing body of research on dynamically adjusting multi-agent workflows according to different task requirements/complexity, including but not limited to [7, 8].
4. The experimental presentation seems rather rushed. At least a few aspects could be analyzed in greater depth: for example, reporting economical cost (LLM API consumption); examining whether, as task complexity increases (e.g., wrt the number of ground-truth documents related to each query), FlashResearch actually learns to allocate different search resources (e.g., varying the number of search nodes); and conducting case studies to analyze the quality of replanning when failed leaf nodes occur (Appendix B seems to omit this discussion).
5. Regarding the utility function U(r) in Line 200, how is it concretely implemented?

---


[1] OWL: Optimized Workforce Learning for General Multi-Agent Assistance in Real-World Task Automation
[2] AgentOrchestra: Orchestrating Hierarchical Multi-Agent Intelligence with the Tool-Environment-Agent(TEA) Protocol
[3] Cognitive Kernel-Pro: A Framework for Deep Research Agents and Agent Foundation Models Training
[4] OAgents: An Empirical Study of Building Effective Agents
[5] https://github.com/ZTE-AICloud/Co-Sight
[6] https://github.com/huggingface/smolagents
[7] FlowReasoner: Reinforcing Query-Level Meta-Agents
[8] Weak-for-Strong: Training Weak Meta-Agent to Harness Strong Executors

**Questions:**

Please refer to Weaknesses

---

> ### Author Response · Authors · 2025-11-24
> **Authors' Response Part 1**
>
> Dear Reviewer YF1V,
>
> Thank you for the thoughtful review and for acknowledging our work's **clean formulation** and **measurable throughput-quality improvements**. We appreciate your detailed feedback on related work and experimental presentation, which has helped us significantly strengthen the paper.
>
> ```
> W1: FlashResearch appears to focus primarily on an offline setting, where data are collected from a local corpus (for instance, the setting in Figure 1 seems to be strongly related to DeepResearchGym). However, I understand that one of the major challenges of deep research actually lies in tool selection and allocation within online web search scenarios—different task types may require different tools (e.g., some can be efficiently solved via a Wikipedia Search API, while others may benefit from Playwright-based visual browsing). Yet, I did not observe any adaptation or discussion regarding this aspect. I understand that DeepResearchGym may only require a single retriever tool, but what kinds of tools does DeepResearchBench actually provide?
> ```
>
> This is an excellent point. We want to clarify a potential misunderstanding about our experimental setup and the broader applicability of our framework.
>
> We use FineWeb as a frozen web snapshot solely to ensure a **reproducible and controlled evaluation** across systems. This follows the recommended protocol in the DeepResearchGym paper. Since our complete evaluation took several days to finish, we needed to eliminate the performance variance caused by temporal web changes. Without a static corpus, comparing systems becomes unreliable because the available information changes between runs.
>
> Meanwhile, DeepResearchBench is a benchmark paper without specifying any particular implementation requirements or tool interfaces. We want to highlight that our framework is **fully compatible with online search** and diverse tool ecosystems. The architecture in Figure 1 shows "*Corpus*" as a generic information source backend. This can be implemented using static web corpora for reproducibility in experiments, live web search APIs like Wikipedia, Google, or Tavily, browser automation tools like Playwright, Selenium, or even specialized local databases and document collections.
>
> Regarding your specific question about tool allocation, we want to emphasize that our core contribution is **orthogonal** to tool selection. We focus on ***the planning and orchestration of the research branches*** regardless of which specific tools are used for conducting the research. Our adaptive planner decides how to decompose queries and allocate computational resources across breadth and depth. Our real-time orchestrator manages when to expand, deepen, or terminate research paths. These mechanisms operate at a higher level of abstraction than tool selection.
>
> That said, we agree that intelligent tool selection and allocation represent an interesting direction for future work. For example, the adaptive planner could be extended to choose appropriate tools based on query characteristics, such as using Wikipedia APIs for factual queries, or code interpreters for computational tasks. The real-time orchestrator could also trigger tool switches if initial attempts fail or yield low-quality results. We will add this discussion to Section 6 to clarify both the generality of our framework and the promising direction of integrating intelligent tool selection.

---

> > ### Author Response · Authors · 2025-11-24
> > **Authors' Response Part 2**
> >
> > ```
> > W2: The paper's survey of current state-of-the-art deep research methods is somewhat insufficient…... there now exist many multi-agent approaches specifically designed for deep research, including but not limited to [1,2,3,4].
> > ```
> >
> > Thank you for highlighting these important concurrent works. We have revised Section 2 to address this gap. We want to highlight FlashResearch's unique focus on real-time, time-budgeted deep research with runtime pruning on tree-structured exploration, which is different from general-purpose multi-agent coordination frameworks. Below, we discuss the core differences between FlashResearch and these works.
> >
> > | Paper | Summary | Key Differences | FlashResearch’s Novelty |
> > | :---- | :---- | :---- | :---- |
> > | OWL \[1\] | A hierarchical multi-agent framework for decoupling domain-agnostic planner (trained via SFT+DPO) from specialized workers for general task automation.  | OWL's RL-optimized planner learns a **static decomposition policy**, which may not adapt well to the query-specific needs in deep research tasks. It employs reactive replanning only on failures. Also, OWL's **coarse-grained worker parallelization** allows workers to operate independently but without fine-grained resource reallocation.  | FlashResearch's adaptive planning adjusts per query. The real-time orchestration enables **fine-grained, dynamic resource reallocation**, which is critical in deep research tasks where information value often becomes clear only mid-research.  |
> > | AgentOrchestra \[2\] | A hierarchical framework with TEA Protocol that unifies tools, environments, and agents. It features a central planner to delegate tasks to sub-agents, and a tool manager agent for dynamic tool creation, retrieval, and reuse. | AgentOrchestra coordinates specialized agents but operates at **agent granularity**. In deep research, the challenge is not just *which agent* to invoke but *which research directions* to pursue within a vast exploration space. AgentOrchestra's **agent-level coordination** cannot efficiently manage the tree-structured exploration in deep research.  | FlashResearch's **tree-based orchestration** with node-level monitoring enables critical capabilities like parallel branch exploration, mid-research evaluation of branch quality, and early termination of redundant paths. These prevent wasted computation on low-value branches. |
> > | Cognitive Kernel-Pro \[3\] | A deep research framework with emphasis on cognitive architecture, memory management, and knowledge integration. It provides abstractions for maintaining a coherent understanding as agents gather information and build knowledge representations across sessions. | Cognitive Kernel-Pro addresses *what* agents know and remember but **not *how efficiently* they explore**. Its memory management does not include specialized mechanisms for *optimizing exploration* in time-constrained deep research tasks.  | FlashResearch **directly tackles the efficiency problem** through early termination, speculative execution, and multi-dimensional parallelization. These mechanisms work together to maximize throughput within time budgets, which are essential for making deep research practical for interactive use.  |
> > | OAgents \[4\] | A modular agent design framework for studying and building multi-agent systems. It provides infrastructure for defining agent roles, configuring communication patterns, and integrating components, with a focus on evaluating design choices across tasks. | OAgents provides a general infrastructure but lacks research-specific optimizations. It focuses more on design modularity and empirical evaluation than on dynamic research tree search.  | FlashResearch introduces **research-specific mechanisms**: adaptive planning that adjusts exploration based on query complexity, real-time orchestration that prunes based on mid-research findings, and speculative execution that reduces latency in tree-structured exploration. These address challenges specific to deep research that general frameworks may fail to solve. |

---

> ### Author Response · Authors · 2025-11-24
> **Authors' Response Part 3**
>
> ```
> W3: In Line 135, the authors claim that most existing agentic workflows 'lack support for real-time replanning.' This statement is inaccurate, as a number of studies have already demonstrated the ability to dynamically adjust plans in real time [5,6]. The authors should take this into account; there is also a growing body of research on dynamically adjusting multi-agent workflows according to different task requirements/complexity, including but not limited to [7,8].
> ```
> Thank you for this important correction. We will revise the statement to acknowledge the relevant prior work.
>
> We examined the codebases of these specific systems \[5, 6\] to better understand their replanning mechanisms. ZTE Co-Sight includes replanning rules within the planner’s system prompt, but its current implementation does not appear to include an automatic replanning trigger.  Smolagents uses a `planning_interval` parameter that triggers replanning at fixed intervals regardless of progress, continuing linearly until `max_steps` is reached. While effective, this interval-based approach may trigger unnecessary replanning even after an answer is found, potentially wasting computation.
>
> In contrast, FlashResearch employs goal-driven orchestration that evaluates research goal satisfaction in real-time and enables early termination once goals are met, reducing unnecessary effort. A brief pairwise comparison is provided below:
>
> |  | smolagents | FlashResearch |
> | :---- | :---- | :---- |
> | Adaptation Strategy | Replanning | Runtime Monitoring \+ Early Termination |
> | Trigger Mechanism | Interval-based | Goal-driven |
> | Efficiency | Potential unnecessary replanning | Terminates early when goals are satisfied |
> | Partial failure handling | May replan entire task | Continues with other branches |
> | Resource Optimization | Fixed `max_steps` limit | Dynamic termination |
> | Parallel Execution | Linear exploration with parallel tool calls | Concurrent tree branch exploration |
>
> We also agree with the reviewer that there is a growing body of work on dynamically adjusting multi-agent workflows according to task requirements and complexity [7,8]. This line of research aligns with our high-level objective of allocating computational resources intelligently across research branches. In the revised paper, we have expanded Section 2 to situate our work more clearly and have highlighted the new discussion in blue color.

---

> > ### Author Response · Authors · 2025-11-24
> > **Authors' Response Part 4**
> >
> > ```
> > W4: The experimental presentation seems rather rushed. At least a few aspects could be analyzed in greater depth: for example, reporting economical cost (LLM API consumption); examining whether, as task complexity increases (e.g., with the number of ground-truth documents related to each query), FlashResearch actually learns to allocate different search resources (e.g., varying the number of nodes explored); and conducting case studies to analyze the quality of replanning when failed leaf nodes occur (Appendix B seems to omit this discussion).
> > ```
> > Thank you for these constructive suggestions. We appreciate your insights on strengthening the experimental analysis. Due to the limited time during the rebuttal period, we conducted several experiments to directly address your concerns about the comprehensiveness of our evaluation:
> >
> > a) Following your advice for an in-depth experiment, we expanded our evaluation to include comprehensive **per-component ablations** that isolate the contribution of each framework component. The [updated Table 1](https://i.ibb.co/wFHqgpHB/updated-table-1.png) now presents these ablation results with 95% confidence intervals, demonstrating how adaptive planning, real-time orchestration, and multi-dimensional parallelization each contribute to the overall system performance.
> >
> > b) To address concerns about the **robustness** of our conclusions **beyond proprietary models**, we conducted additional experiments using the open-source `Qwen3-235B-A22B-Instruct-2507` model. We report the results in [Table 6 of the updated paper](https://i.ibb.co/5X37x9QS/table-6.png) with 95% confidence intervals.
> >
> > Despite the lower throughput due to the higher latency of locally-served `Qwen3-235B-A22B-Instruct-2507`, FlashResearch consistently outperforms the baseline GPT-Researcher system, delivering 2.77x and 4.51x throughput improvements in the 2-minute and 10-minute setups, respectively. Notably, the overall quality achieved by FlashResearch with 2-minute execution (84.91 ± 0.76) surpasses the 10-minute GPT-Researcher (83.48 ± 0.79), demonstrating a 5x speed-up consistent with our main experiments in Table 1 using proprietary LLMs.
> >
> > Examining the submetrics reveals interesting patterns: the 2-minute executions achieve higher clarity, depth, and breadth scores than the 10-minute executions, potentially due to the open-source model’s incapabilities in managing a long-context window for effective information aggregation. However, 10-minute executions significantly outperform in terms of support and citation recall, indicating that the longer execution still benefits the faithfulness and evidentiary support of the research outputs. These results further validate that FlashResearch's architectural benefits transfer across different model families and deployment scenarios.
> >
> > **Below, we continue with our response to W4.**

---

> ### Author Response · Authors · 2025-11-24
> **Authors' Response Part 5**
>
> Response to W4 (continued):
> ```
> W4: The experimental presentation seems rather rushed. At least a few aspects could be analyzed in greater depth: for example, reporting economical cost (LLM API consumption); examining whether, as task complexity increases (e.g., with the number of ground-truth documents related to each query), FlashResearch actually learns to allocate different search resources (e.g., varying the number of nodes explored); and conducting case studies to analyze the quality of replanning when failed leaf nodes occur (Appendix B seems to omit this discussion).
> ```
> c) In response to your request for **economic cost analysis**, we have conducted a comprehensive profiling of LLM API consumption patterns. Due to space constraints in the main paper, we present these results in [Table 7 of the updated paper](https://i.ibb.co/N6hPH3fk/table-7.png).
>
> d) To address your question about whether FlashResearch **adapts its resource allocation based on task complexity,** we want to highlight that in the case studies in Appendix B, we indeed find patterns that queries on broader topics can result in a wider, deeper tree structure, and narrow, domain-specific query yields a more compact tree. Since we use FineWeb as the search corpus, there are no “ground truth” documents for each query. Therefore, in addition to the qualitative case analysis, we add a **quantitative analysis examining success and failure patterns** across different query types.
>
> Overall, FlashResearch achieves higher quality scores on 70% of the questions on DeepResearchGym under 10-minute time budget. For the two groups of questions where FlashResearch either outperforms (70 questions) or underperforms (30 questions) the baseline, we examine the patterns based on query characterizations from the original Researchy Questions dataset. We compute average scores for each group across different feature dimensions, as visualized in [Figure 4 of the updated paper](https://i.ibb.co/s9HZN10Z/failure-analysis.png).
>
> As shown in the leftmost subfigure, the questions where FlashResearch “wins” tend to score higher on the “*knowledge-intensive*” and “*reasoning-intensive*” dimensions.
>
> Looking at the quality submetrics, FlashResearch achieves better “*Clarity*”, “*Depth*”, “*Balance*”, and “Breadth” on the questions that are more “*multi-faceted*”, “*knowledge-intensive*”, and “*reasoning-intensive*”. This observation aligns directly with our core claims: adaptive planning and real-time orchestration enable the system to maintain a robust tradeoff between expansive coverage and focused analysis, especially for  “*multi-faceted*” questions that demand breadth expansion, and “*knowledge-intensive*” and “*reasoning-intensive*” questions that can benefit from more in-depth research within the same budget.
>
> In contrast, the questions where FlashResearch underperforms exhibit higher average scores in “*ambiguous*” and “*incompleteness*” dimensions. This suggests an important limitation that when a question has multiple interpretations, or when its intent is unclear due to missing context, the adaptive planner and real-time orchestrator in FlashResearch may struggle to make the right decisions, leading to suboptimal performance.
>
> **Below, we continue with our response to W4.**

---

> ### Author Response · Authors · 2025-11-24
> **Authors' Response Part 6**
>
> Response to W4 (continued):
> ```
> W4: The experimental presentation seems rather rushed. At least a few aspects could be analyzed in greater depth: for example, reporting economical cost (LLM API consumption); examining whether, as task complexity increases (e.g., with the number of ground-truth documents related to each query), FlashResearch actually learns to allocate different search resources (e.g., varying the number of nodes explored); and conducting case studies to analyze the quality of replanning when failed leaf nodes occur (Appendix B seems to omit this discussion).
> ```
> f) Regarding analysis of replanning after failed leaf nodes, we checked the execution logs of the example query shown in our updated paper's Figure 7. The user query explicitly requests *"only cites high-quality, English-language journal articles"*. Therefore, when the orchestrator detects that the node does not satisfy the research goal, it terminates the branch early to avoid further costs. In this specific case, the orchestrator's output rationale was as follows:
>
> >*The current research progress mainly comprises news articles, reports, and policy discussions rather than high-quality, peer-reviewed English journal articles specifically focused on AI-driven labor market restructuring. While the materials are relevant to the broader topic, they do not meet the stringent criteria of academic journal quality necessary to fully satisfy the research goal. The findings provide useful context and cover significant aspects of AI's impact on the labor market, but they lack the depth, methodological rigor, and academic sourcing expected from high-quality journal articles. Consequently, although there is moderate progress on the subject, key elements of the goal remain unmet, suggesting that it would be more cost-efficient to redirect or deepen the research efforts rather than continuing on the current path.*
>
> When the initial research plan failed to yield the requested high-quality journal articles, the orchestrator recognized this mismatch and terminated the low-value path, thereby freeing computational resources for alternative exploration strategies. We will add this to the discussion in Appendix B.

---

> > ### Author Response · Authors · 2025-11-24
> > **Authors' Response Part 7**
> >
> > ```
> > W5: Regarding the utility function U(r) in Line 200, how is it concretely implemented?
> > ```
> > We appreciate this question as it helps clarify an important aspect of our formalization. The utility function $U(r)$ in Equation 5 serves as an abstract representation of the evaluation function that measures the quality of the final response, *i.e.*, the deep research report generated by the system.
> >
> > In our experiments, the utility function is implemented through the standard LLM-as-a-judge protocols established by each benchmark, as described in Section 5.1. For DeepResearchGym, we follow the benchmark's recommended practice using GPT-4.1-mini with detailed rubrics covering Quality, Relevance, and Faithfulness dimensions. For DeepResearch Bench, we employ the Gemini-2.5 family models to compute RACE and FACT metrics. The complete evaluation protocols, including specific prompts and scoring procedures, are detailed in the respective benchmark papers.
> >
> > By treating $U(r)$ abstractly in our formulation, FlashResearch remains agnostic to the specific quality assessment mechanism. In principle, $U(r)$ could also be instantiated with human evaluations or learned reward functions. The deliberate separation highlights that FlashResearch's core contributions of improving efficiency and throughput can operate independently of how quality is ultimately evaluated.

---

### Official Review · Reviewer_S2LH · 2025-10-31

**Soundness:** 3
**Presentation:** 3
**Contribution:** 3
**Rating:** 4
**Confidence:** 2

**Summary:**

The paper proposes FlashResearch, a framework for deep research. It introduces an adaptive planner for reasoning depth, a real-time orchestration layer that dynamically adjusts compute allocation, and a multi-dimensional parallelization scheme that explores both the breadth and depth of reasoning in parallel. On benchmark datasets, the framework achieves strong accuracy while improving inference efficiency.

**Strengths:**

1: The paper is clearly written and tackles an important efficiency problem in deep research; the method and experiments are well aligned with the motivation.

2: By integrating adaptive planning, real-time orchestration, and multi-dimensional parallelization, the system simultaneously boosts throughput and reduces latency, making it readily deployable in production.

**Weaknesses:**

1: Both the adaptive planner and the real-time orchestration policies rely on LLM judgments to decide when to expand, deepen, or terminate paths. This makes the decision process hard to interpret.
2: Evaluating the goal-satisfaction level and quality score of every node requires on-the-fly calls to an LLM, incurring measurable extra overhead.
3: The choice of 2-minute and 10-minute time budgets is motivated by prior human–computer interaction findings, but the paper lacks a broader sensitivity analysis across time budgets and does not discuss external/API-induced latency that could waste the budget. This limits understanding of robustness under different deployment conditions.

**Questions:**

See weakness

---

> ### Author Response · Authors · 2025-11-21
> **Authors' Response Part 1**
>
> Dear Reviewer S2LH,
>
> Thank you for your thoughtful review and for recognizing that our paper **addresses an important efficiency problem in deep research**. We address each of your concerns below.
> ```
> W1: Both the adaptive planner and the real-time orchestration policies rely on LLM judgments to decide when to expand, deepen, or terminate paths. This makes the decision process hard to interpret.
> ```
> We acknowledge this valid concern about interpretability. Our formulations (Equations 7, 8, and 9\) are fundamentally **policy-agnostic**. The value of our agent orchestration framework, including adaptive planning, speculative execution, mid-research pruning, and multi-dimensional parallelism, exists independent of how these policies are implemented.
>
> Due to the complexity of deep research tasks, the planning and orchestration require an in-depth assessment of exponentially growing context from dimensions like information quality, coverage gaps, and relevance, which are hard to manually annotate and difficult to capture with rule-based heuristics. In contrast, LLM policies naturally adapt to diverse query types without requiring domain-specific rules for each category.
>
> Also, we want to highlight that our current implementation using **thinking models** (o3-mini) provides natural language justifications to ease users’ interpretation. Importantly, these models provide **explicit reasoning traces**, enabling interpretation of decisions and error tracking. For example, when the orchestrator decides to terminate a node in the example of our paper’s Figure 7, it explains the rationale: since the user explicitly requests *"only cites high-quality, English-language journal articles"*, the orchestrator detects that the node does not satisfy the research goal and terminates early to avoid further costs. This transparency already facilitates debugging and system improvement.
>
> > *The current research progress mainly comprises news articles, reports, and policy discussions rather than high-quality, peer-reviewed English journal articles specifically focused on AI-driven labor market restructuring. While the materials are relevant to the broader topic, they do not meet the stringent criteria of academic journal quality necessary to fully satisfy the research goal. The findings provide useful context and cover significant aspects of AI's impact on the labor market, but they lack the depth, methodological rigor, and academic sourcing expected from high-quality journal articles. Consequently, although there is moderate progress on the subject, key elements of the goal remain unmet, suggesting that it would be more cost-efficient to redirect or deepen the research efforts rather than continuing on the current path.*
>
> ```
> W2: Evaluating the goal-satisfaction level and quality score of every node requires on-the-fly calls to an LLM, incurring measurable extra overhead.
> ```
>
> Thank you for raising this important consideration. We conducted a profiling analysis of our 10-minute budgeted experiment on DeepResearchGym. As listed in the table, the extra overhead from orchestration only accounts for **7.21% of the total LLM API costs**. At the same time, in total of 32.12 nodes are pruned during the research progress, accounting for an **average savings of \\$0.1665**, which is **3.39 times** more than the orchestration costs of \\$0.0491 per question.
>
> | Avg. of 100 Questions | 10 mins |
> | :---- | :---- |
> | Input Tokens for Orchestration | 25646 |
> | Output Tokens for Orchestration | 4756 |
> | Costs for Orchestration | \\$0.0491 |
> | Total Costs per Question | \\$0.6811 |
> |  |  |
> | \# Pruned Nodes | 32.12 |
> | \# Total Planning \+ Research Nodes | 131.42 |
> | Average Costs per Node | $0.0052 (\\$0.6811/131.42) |
> | Total Savings from Orchestration | \\$0.1665 (32.12\*\\$0.0052) |

---

> > ### Author Response · Authors · 2025-11-21
> > **Authors' Response Part 2**
> >
> > ```
> > W3: The choice of 2-minute and 10-minute time budgets is motivated by prior human-computer interaction findings, but the paper lacks a broader sensitivity analysis across time budgets and does not discuss external-API-induced latency that could waste the budget.
> > ```
> > The 2-minute and 10-minute thresholds were chosen based on established HCI research on cognitive flow and task switching. Following your comments, we conducted a **sensitivity analysis on time budgets** via **independent samples t-tests**. As shown in [Table 5 in the updated paper](https://i.ibb.co/VY30tMvt/sensitivity-test.png), extending time budgets from 2 to 10 minutes generally improves performance across models, with consistent significant gains in overall quality (mean change Δ ranges from \+3.8 to \+5.5, all p\<0.001).
> >
> > In particular, most robust improvements occur in *Support* (Δ \+16-18 across models, all highly significant with p=0.0000), suggesting more time allows for better information gathering, citation integration, and backing of claims in the final report. *Breadth* and *Balance* improve significantly in FlashResearch and ablated FlashResearch\*, showing our system’s superiority in effectively allocating the resources with additional time budget.
> >
> > Overall, the baseline GPT-Researcher shows the broadest sensitivity to time increases, with 8/10 metrics significantly improved. This reflects the baseline system’s heavy reliance on raw time for thorough evidence gathering, precision, and depth, making it less efficient under tight time constraints. In comparison, FlashResearch displays balanced yet targeted sensitivity (significant in 6/10 metrics), performing efficiently at 2 minutes while using extra time strategically.
> >
> >
> > In addition, we analyzed the latency composition in our DeepResearchGym experiments and found that each LLM call takes an average of 20.776s of wall time, while each FineWeb search takes 2.328s. With a typical 10-minute run involving \>600 LLM calls and 500-600 web retrieval operations, cumulative API latency could reach up to **23×** the time budget if executed sequentially.
> >
> > However, FlashResearch's **multi-dimensional parallelization** effectively mitigates this bottleneck. The system issues concurrent API calls across multiple research nodes, **overlaps I/O wait times with computation**, and maintains high resource utilization. Critically, the asynchronous architecture ensures that **slow API responses in one branch do not block progress in other branches**—an advantage over both sequential execution (where each call blocks subsequent work) and layer-wise parallelization (where slow nodes delay entire layers). By enabling fully asynchronous exploration across breadth and depth dimensions, FlashResearch transforms what would be a severe latency bottleneck into manageable overhead distributed across concurrent execution paths.

---

### Official Review · Reviewer_Yz8E · 2025-10-31

**Soundness:** 3
**Presentation:** 3
**Contribution:** 3
**Rating:** 6
**Confidence:** 3

**Summary:**

Given the limited runtime adaptability of existing methods for deep research tasks, this paper introduces a dynamic planner designed to adjust the breadth and depth of the research tree in real time. The planner leverages utility-guided strategies, where utility is determined by LLM-based judges. The proposed approach is evaluated across multiple deep research benchmarks, showing improved accuracy and reduced latency.

**Strengths:**

1. The examples provided in Section 3.2 highlight the necessity and potential benefits of improving the configuration of breadth and depth.
2. The explicit formulation of controlling the research tree's breadth and depth is both logically sound and empirically effective.
3. The paper is well-written and easy to understand.

**Weaknesses:**

1. Because the method relies on the current implementation of LLM-as-a-judge, its effectiveness depends heavily on the LLM’s ability to make accurate judgments.
2. The evaluation is limited to a single model family (Gemini-2.5); testing with additional models, such as smaller open-source ones, would strengthen the results.

**Questions:**

1. Can the proposed approach be generalized to utility models beyond LLM-as-a-judge?
2. Given the asynchronous operations and increased number of explored nodes, does this result in higher computational or operational costs?

---

> ### Author Response · Authors · 2025-11-21
> **Authors' Response Part 1**
>
> Dear Reviewer Yz8E,
>
> Thank you for highlighting our contributions in providing **a clear and sound formulation of the tree-structured deep research tasks**. Below, we provide our detailed response, addressing each of your raised weaknesses and questions.
>
> ```
> W1: Because the method relies on the current implementation of LLM-as-a-judge, its effectiveness depends heavily on the LLMs ability to make accurate judgments.
> ```
> ```
> Q1: Can the proposed approach be generalized to utility models beyond LLM-as-a-judge?
> ```
> We wish to clarify two distinct uses of LLMs in our work:
>
> **LLMs for evaluation:** If you are concerned about the reliability of “LLM-as-a-judge” for evaluation, we want to highlight that we follow the **standard evaluation protocol established by both DeepResearchGym and DeepResearch Bench**. The LLM-as-a-judge approach is widely adopted in the research community due to difficulties in human evaluation of deep research reports, which are typically long and require domain expertise to assess.
>
> To address concerns about evaluation reliability, we conducted a rigorous **inter-rater reliability analysis**. We reran the LLM judge five times and calculated the **Intraclass Correlation Coefficient (ICC)** to measure consistency across runs. As shown in [Table 4 in the updated paper](https://i.ibb.co/CsYbjw71/icc-test.png), ICC scores are consistently above 0.85, with most exceeding 0.9, indicating excellent reliability and consistency of the LLM judge across multiple runs on various metrics.
>
> **LLMs for planning and orchestration:** Regarding the use of LLMs in our adaptive planner and real-time orchestrator, we want to emphasize that our formulations in Equations 7, 8, and 9 are fundamentally **policy-agnostic**.
>
> To directly answer your question: Yes, the approach generalizes beyond LLM-based policies. The value of our agent orchestration framework, including adaptive planning, speculative execution, mid-research pruning, and multi-dimensional parallelism, **exists independent of how these policies are implemented**.
>
> For example, there exist multiple alternative implementations:
> i) Learned policies via reinforcement learning. The policies could be optimized to maximize quality-cost ratios through RLHF training on well-curated reward functions.
> ii) Heuristic-based utility models. Decisions could be driven by statistical metrics such as TF-IDF coverage for determining breadth (like how many distinct aspects remain unexplored); information gain thresholds for depth decisions (when marginal utility drops below a threshold); and topic overlap for termination.
>
> We had also considered training separate models for decision-making. However, collecting human annotations for deep research orchestration decisions is extremely difficult given the task complexity. Annotators would need to assess whether a partial research path should be expanded, deepened, or terminated, demanding a deep understanding of both the query and the exponentially growing context. Using LLM-generated labels as a proxy reward function to train separate models would essentially replicate the current LLM-based approach with added complexity.
>
> Also, we want to highlight that our current implementation using **thinking models** (o3-mini) has been shown to be effective in experiments. Importantly, these models provide **explicit reasoning traces**, enabling interpretation of decisions and error tracking. For example, when the orchestrator decides to terminate a node in the question example of our Figure 7, it explains the rationale: since the user explicitly requests *"only cites high-quality, English-language journal articles"*, the orchestrator detects that the node does not satisfy the research goal and terminates early to avoid further costs. This transparency already facilitates debugging and system improvement.
>
> > *The current research progress mainly comprises news articles, reports, and policy discussions rather than high-quality, peer-reviewed English journal articles specifically focused on AI-driven labor market restructuring. While the materials are relevant to the broader topic, they do not meet the stringent criteria of academic journal quality necessary to fully satisfy the research goal. The findings provide useful context and cover significant aspects of AI's impact on the labor market, but they lack the depth, methodological rigor, and academic sourcing expected from high-quality journal articles. Consequently, although there is moderate progress on the subject, key elements of the goal remain unmet, suggesting that it would be more cost-efficient to redirect or deepen the research efforts rather than continuing on the current path.*
>
> To summarize, we view LLM-based policies as a strong initial implementation that demonstrates the framework's potential. We acknowledge this as important future work and will clarify the policy-agnostic nature of our framework more explicitly in the paper.

---

> > ### Author Response · Authors · 2025-11-21
> > **Authors' Response Part 2**
> >
> > ```
> > W2: The evaluation is limited to a single model family (Gemini-2.5), testing with additional models, such as smaller open-source ones, would strengthen the results.
> > ```
> > We appreciate this feedback and wish to clarify a misunderstanding about our experimental setup: our evaluation is conducted on **GPT models (gpt-4.1-mini)** for DeepResearchGym, and the **Gemini-2.5 family** for DeepResearch Bench. These follow the benchmarks’ standard protocol, not our choice.
> >
> > If you are referring to the operating models for deep research execution, as stated in Appendix A.3, we use GPT-4.1-mini-2025-04-14 for the main research processing, and  o3-mini-2025-01-31 for adaptive research planning and real-time orchestration.
> >
> > We want to highlight that our contribution mainly comes from the **architectural efficiency** rather than the models. The **4.11x throughput gain** is achieved through parallelization and early termination, which should benefit any deep research system by enabling any model to explore more research paths and gather more evidence within the budget, regardless of the operating models.
> >
> > We agree that experiments with additional models will enhance the comprehensiveness of our evaluation. We are running experiments on the open-source `Qwen3-235B-A22B-Instruct-2507` (an MoE model with 22B activated parameters) and will share the results here as soon as they are available.
> >
> > ```
> > Q2: Given the asynchronous operations and increased number of explored nodes, does this result in higher computational or operational costs?
> > ```
> >
> > Our evaluation uses **fixed time budgets** (2 or 10 minutes) as the constraint. Every system runs for the same duration; the key difference lies in how that time is utilized. FlashResearch employs multi-dimensional parallelization with speculative execution to minimize idle time, enabling exploration of 98.43 nodes in 10 minutes, 4.11x more than the baseline’s 23.12 nodes.
> >
> > In other words, if we use the research amount as the fixed constraint, **FlashResearch can finish faster without extra operational costs**. Achieving equivalent research coverage with the baseline system would require the same token cost and significantly more execution time.
> >
> > Therefore, we believe cost-effectiveness should be evaluated as *quality per dollar*. Our token-usage analysis shows that a **2-minute FlashResearch** **run costs $0.2071** on average (155,409 input; 10,031 output tokens; mixture of gpt-4.1-mini and o3-mini) and already outperforms a **10-minute GPT-Researcher run costing $0.3941** (299,093 input; 18,432 output tokens). Despite the orchestration overhead, FlashResearch delivers better performance per unit cost due to early termination and adaptive planning, which concentrate computation on high-value paths rather than wasting resources on saturated or unproductive branches.

---

> > > ### Author Response · Authors · 2025-11-26
> > > **Updated Experiment Results on Additional Open-source Model**
> > >
> > > Following your advice about **testing on smaller, open-source models**, we conducted additional experiments using the open-source `Qwen3-235B-A22B-Instruct-2507` model (an MoE model with 22B activated parameters). We report the results in [Table 6 of the updated paper](https://i.ibb.co/5X37x9QS/table-6.png) with 95% confidence intervals.
> > >
> > > Despite the lower throughput due to the higher latency of locally-served `Qwen3-235B-A22B-Instruct-2507`, FlashResearch consistently outperforms the baseline GPT-Researcher system, delivering **2.77x and 4.51x throughput improvements** in the 2-minute and 10-minute setups, respectively. Notably, the overall quality achieved by FlashResearch with 2-minute execution (84.91 ± 0.76) surpasses the 10-minute GPT-Researcher (83.48 ± 0.79), demonstrating a **5x speed-up** consistent with our main experiments in Table 1 using proprietary LLMs.
> > >
> > > Examining the submetrics reveals interesting patterns: the 2-minute executions achieve higher *clarity*, *depth*, and *breadth* scores than the 10-minute executions, potentially due to the open-source model’s incapabilities in managing a long-context window for effective information aggregation. However, 10-minute executions significantly outperform in terms of *support* and *citation recall*, indicating that the longer execution still benefits the faithfulness and evidentiary support of the research outputs. These results further validate that FlashResearch's architectural benefits transfer across different model families and deployment scenarios.
> > >
> > > We hope these additional experiments and analyses help clarify your concerns. If you have any other questions or wish to discuss further, please feel free to reach out. We are looking forward to the opportunity to continue the discussion.

---

### Official Review · Reviewer_ZoH6 · 2025-11-01

**Soundness:** 3
**Presentation:** 3
**Contribution:** 2
**Rating:** 4
**Confidence:** 3

**Summary:**

This paper presents the FlashResearch framework, designed to enable efficient and scalable deep research. The framework operates in parallel, dynamically decomposing complex queries into subtasks at runtime. An adaptive planning module determines the appropriate breadth and depth of exploration, while a real-time orchestration layer monitors progress and optimizes research pathways for maximum efficiency.

**Strengths:**

- The paper is clearly written and well-structured.
- Demonstrates improved research quality under fixed computational budgets, achieving up to a 5× speedup.

**Weaknesses:**

- The proposed framework is evaluated on only two benchmarks, with relatively small sample sizes (100 and 50 examples). The rationale for not using the full datasets is unclear. Evaluating on larger datasets or additional benchmarks would strengthen the empirical validity of the results.
- The use of LLMs for evaluation raises concerns regarding metric reliability and potential bias. This should be discussed in greater detail.
- Comparisons are limited to GPT-Researcher. Including experiments with additional models would improve the robustness and generalizability of the findings.
- The performance results are mixed, and further analysis is needed to clarify the conditions under which the proposed method is most effective.
- The paper lacks an ablation study, which would help isolate and assess the contribution of individual components within the framework.

**Questions:**

Was any error analysis performed to better understand the sources of the model’s successes and failures?

---

> ### Author Response · Authors · 2025-11-21
> **Authors' Response Part 1**
>
> Dear Reviewer ZoH6,
>
> Thank you very much for your insightful comments and for highlighting the value of our work in **achieving substantial deep research speedup**. We carefully address your raised weaknesses and questions below.
>
> ```
> W1: The proposed framework is evaluated on only two benchmarks, with relatively small sample sizes (100 and 50 examples). The rationale for not using the full datasets is unclear. Evaluating on larger datasets or additional benchmarks would strengthen the empirical validity of the results.
> ```
>
> We understand your concerns about the generalization ability of our framework, which is precisely why we evaluated our approach on two different benchmarks.
>
> On **DeepResearch Bench**, we evaluated the **entire 50 English questions** **without sampling**.
>
> On **DeepResearchGym, we randomly sampled 100 queries** to manage computation costs (each system requires 2-10 minutes per query, yielding 600 total runs across 3 systems × 2 time budgets \= 20-100 compute hours; scaling to 1000 queries will require 200-1000 compute hours).
>
> To address your concerns about the representativeness of our 100-sample, we conducted rigorous statistical tests comparing our sample distribution to the full benchmark’s distribution. DeepResearchGym's questions are drawn from the Researchy Questions dataset, which provides detailed characterizations across multiple dimensions. We performed 2 non-parametric tests and an equivalence test to verify distributional equivalence:
>
> * **Mann-Whitney U test:** Tests whether samples of two groups have the same distribution
> * **Kolmogorov-Smirnov test:** Tests maximum distance between cumulative distributions
> * **Equivalence Testing with Two One-Sided Tests:** Tests whether two groups are equivalent by a predefined margin.
>
> As shown in [Table 3 in the updated paper](https://i.ibb.co/27M8FhKS/sampling-test.png), across all feature dimensions, both non-parametric tests consistently show p-values \>0.05, indicating no significant distributional differences between the 100 sampled set and the 1000 DeepResearchGym question set. Furthermore, TOST shows p-values \<0.05 across all 10 dimensions at the medium effect size threshold (δ=0.5×SD), with Cohen's d values ranging from \-0.234 to 0.070.
>
> These statistical validations indicate that **the sampled set is representative for generalizing results to the full DeepResearchGym dataset**. We will add this discussion to Section 5.1 and our detailed analysis to the appendix.
>
> ```
> W2: The use of LLMs for evaluation raises concerns regarding metric reliability and potential bias. This should be discussed in greater detail.
> ```
>
> We understand that this is an important concern. Due to the task complexity of deep research, where the output reports are usually long and require domain expertise to understand, it is really difficult to evaluate the reports manually. Therefore, LLM-as-a-judge is the **standard evaluation protocol established by both DeepResearchGym and DeepResearch Bench**, and their LLM-judge rubrics are shown in their original papers to have a high level of alignment with human users’ judgment (an average pairwise Cohen’s κ score of 0.87 in DeepResearchGym, and an overall Pearson Correlation of 99.54 in DeepResearch Bench).
>
> We agree that LLM judges may display systematic biases, such as favoring outputs from their own model family over those from other models. To mitigate this, in our experiments, we used diverse judges, i.e., **GPT-4.1-mini** for DeepResearchGym, and **Gemin 2.5-family** for DeepResearch Bench.
>
> To further address your concerns, we rerun Table 1’s evaluation 5 times and conduct a rigorous statistical analysis.  Specifically, to measure the reliability across multiple LLM-judge runs, we calculate the **Intraclass Correlation Coefficient (ICC)**. As shown in [Table 4 in the updated paper](https://i.ibb.co/CsYbjw71/icc-test.png), the ICC scores are primarily \>0.9 (all \>0.85), indicating a reliable and consistent agreement across multiple runs and various metrics of the LLM judge. We will add this discussion to Section 5 and our analysis results to the appendix.
>
> We also update [Table 1](https://i.ibb.co/wFHqgpHB/updated-table-1.png), reporting the mean and 95% confidence intervals for all metrics, with more extensive ablation studies.

---

> ### Author Response · Authors · 2025-11-21
> **Authors' Response Part 2**
>
> ```
> W3: Comparisons are limited to GPT-Researcher. Including experiments with additional models would improve the robustness and generalizability of the findings.
> ```
> While we appreciate the reviewer’s suggestion, we would like to clarify our positioning. As also recognized by reviewers Yz8E, S2LH, and YF1V, the primary contribution of our work is the formulation of deep research as a tree-structured optimization problem, along with three orchestration components that significantly improve runtime efficiency. These mechanisms are **model-agnostic** and applicable to various deep research systems.
>
> Empirically, we restricted our comparison to GPT-Researcher to **ensure a controlled and fair evaluation** in which performance differences can be attributed solely to our orchestration mechanisms. Deep research is a complex task that involves multiple modules, including planning, web search, retrieval, reasoning, summarization, context aggregation, and report generation. Since our contributions focus specifically on planning and orchestration, using a single, stable baseline allows us to **isolate their impact while keeping all other modules fixed** (e.g., web search, prompting templates).
>
> However, we agree that experimenting with additional models will enhance the robustness of our evaluation. We are running additional experiments on the open-source `Qwen3-235B-A22B-Instruct-2507` (an MoE model with 22B activated parameters) and will share the results here as soon as they are available.
>
> ```
> W4: The performance results are mixed, and further analysis is needed to clarify the conditions under which the proposed method is most effective.
> ```
> ```
> Q1: Any error analysis performed to better understand the sources of the model's successes/failures?
> ```
>
> To address your concerns, we conduct a detailed analysis of the success and failure patterns in which FlashResearch either outperforms or underperforms the baseline.
>
> Overall, FlashResearch achieves higher quality scores on 70% of the questions on DeepResearchGym under the 10-minute time budget. Then, for the two separate groups of questions on which FlashResearch *“wins”* (70 questions) or *“loses”* (30 questions), we examine the group patterns based on the characterizations from the original Researchy Questions dataset (see Figure 7 in the Researchy Questions paper for definitions).
>
> In [Figure 4 of the updated paper](https://i.ibb.co/s9HZN10Z/failure-analysis.png), we plot the **average characterization scores** for the questions where FlashResearch *wins* or *loses*, based on the score dimensions (as in the x-axis), and across different
>  quality evaluation metrics (as in the subfigure title).
>
> As shown in the leftmost subfigure, overall, the questions where FlashResearch “wins” tend to score higher on the “*knowledge-intensive*” and “*reasoning-intensive*” dimensions.
>
> Looking at the quality submetrics, FlashResearch achieves better “*Clarity*”, “*Depth*”, “*Balance*”, and “*Breadth*” on the questions that are more “*multi-faceted*”, “*knowledge-intensive*”, and “*reasoning-intensive*”. This aligns with the findings in the paper that adaptive planning and real-time orchestration enable our system to maintain a robust tradeoff between expansive coverage and focused analysis, especially for  “*multi-faceted*” questions that demand breadth expansion, and “*knowledge-intensive*” and “*reasoning-intensive*” questions that can benefit from more in-depth research within the same budget.
>
> In contrast, the questions where FlashResearch “*loses*” have higher average scores in “*ambiguous*” and “*incompleteness*” dimensions. This suggests that when a question has multiple interpretations, or when its intent is unclear due to missing context, the adaptive planner and real-time orchestrator in FlashResearch may struggle to make the right decisions, leading to suboptimal performance.
>
> ```
> W5: The paper lacks an ablation study, which would help isolate and assess the contribution of individual components within the framework.
> ```
>
> **We already include an ablated version** of FlashResearch, reported in Tables 1 and 2 as `FlashResearch*`. To further isolate the contribution of individual components, as you suggested, we expand this to per-component breakdowns, as shown in the updated [Table 1](https://i.ibb.co/wFHqgpHB/updated-table-1.png), where we also report the 95% confidence interval.

---

> ### Author Response · Authors · 2025-11-26
> **Updated Experiment Results on Additional Open-source Model**
>
> Following your suggestions about **experimenting with additional models**, we conducted additional experiments using the open-source `Qwen3-235B-A22B-Instruct-2507` model (an MoE model with 22B activated parameters). We report the results in [Table 6 of the updated paper](https://i.ibb.co/5X37x9QS/table-6.png) with 95% confidence intervals.
>
> Despite the lower throughput due to the higher latency of locally-served `Qwen3-235B-A22B-Instruct-2507`, FlashResearch consistently outperforms the baseline GPT-Researcher system, delivering **2.77x and 4.51x throughput improvements** in the 2-minute and 10-minute setups, respectively. Notably, the overall quality achieved by FlashResearch with 2-minute execution (84.91 ± 0.76) surpasses the 10-minute GPT-Researcher (83.48 ± 0.79), demonstrating a **5x speed-up** consistent with our main experiments in Table 1 using proprietary LLMs.
>
> Examining the submetrics reveals interesting patterns: the 2-minute executions achieve higher *clarity*, *depth*, and *breadth* scores than the 10-minute executions, potentially due to the open-source model’s incapabilities in managing a long-context window for effective information aggregation. However, 10-minute executions significantly outperform in terms of *support* and *citation recall*, indicating that the longer execution still benefits the faithfulness and evidentiary support of the research outputs. These results further validate that FlashResearch's architectural benefits transfer across different model families and deployment scenarios.
>
> We hope these additional experiments and analyses may address your concerns. If you have any further questions or would like to discuss our response in more detail, please feel free to reply. We will be happy to continue the discussion with you.

---

### Author Response · Authors · 2025-12-03
**Summary of Rebuttal**

We are thankful to the reviewers for their constructive feedback and recognition of our work. We are encouraged that the reviewers found FlashResearch to be **"clearly written and well-structured"** (`Reviewers ZoH6, S2LH`), tackling an **"important efficiency problem"** (`Reviewer S2LH`). Specifically, the formulation is a **"clean, asynchronous tree abstraction"** (`Reviewer YF1V`) that is **"logically sound and empirically effective"** (`Reviewer Yz8E`). Reviewers also highlighted the practical impact, noting the method demonstrates **"measurable throughput-quality lift"** (`Reviewer YF1V`) and **"up to a 5x speedup"** (`Reviewer ZoH6`) that can be **readily deployable in production** (`Reviewer S2LH`).

In response to the reviews, we conducted extensive additional experiments and analyses. Our response focuses on three key aspects: establishing evaluation reliability, showing generalizability to open-source LLM, and deepening the analysis of costs and components.

### **a) Robustness and Reliability of Evaluation**

Common concerns were raised regarding the reliability of LLM-as-a-judge metrics (`Reviewers ZoH6, Yz8E`) and the sample size of the benchmarks (`Reviewer ZoH6`). We conducted several statistical analyses accordingly:

* **Reliability of LLM-as-a-judge:** We conducted a rigorous statistical analysis of the LLM judges. As reported in [the new Table 4](https://i.ibb.co/CsYbjw71/icc-test.png), the Intraclass Correlation Coefficient (ICC) scores are consistently **\>0.85** across multiple runs, confirming that the LLM judges' evaluations are highly reliable and consistent.
* **Sample Representativeness:** We performed statistical validation (Mann-Whitney U, Kolmogorov-Smirnov, and Equivalence Testing) comparing our 100-sample subset to the full DeepResearchGym dataset. Results in [the new Table 3](https://i.ibb.co/27M8FhKS/sampling-test.png) confirm no significant distributional differences, validating the empirical soundness of our evaluation.

### **b) Generalizability to Open-Source LLM**

Both `Reviewers ZoH6, Yz8E` suggested evaluating beyond the proprietary LLMs to prove robustness. Therefore, we validated FlashResearch on the open-source **Qwen3-235B-A22B-Instruct**. As shown in [the new Table 6](https://i.ibb.co/5X37x9QS/table-6.png), FlashResearch consistently outperforms the baseline, delivering **2.77x** (2-min) and **4.51x** (10-min) **throughput improvements**. Notably, our 2-minute execution with Qwen achieved higher quality (84.91) than the baseline's 10-minute run (83.48), showing a **5x speed-up** consistent to our proprietary LLM experiments.

### **c) Comprehensive Experimental Analysis**

Reviewers requested deeper insights into component contributions (`Reviewer ZoH6`), economic costs (`Reviewers S2LH, YF1V`), and failure modes (`Reviewer ZoH6`). Correspondingly, we conducted comprehensive analyses, including:

* **Expanded Ablation Studies:** We updated [Table 1](https://i.ibb.co/wFHqgpHB/updated-table-1.png) to include per-component breakdowns with 95% confidence intervals, isolating the specific gains from adaptive planning and real-time orchestration.
* **Economic & Latency Profiling:** We provided a detailed cost analysis in [the new Table 7](https://i.ibb.co/N6hPH3fk/table-7.png), showing that orchestration overhead is negligible (\~7% of costs) compared to the savings from pruning low-value paths. We also added a sensitivity analysis in [the new Table 5](https://i.ibb.co/VY30tMvt/sensitivity-test.png), validating FlashResearch's superior resource allocation as time budgets increase.
* **Failure Pattern Analysis:** We added [Figure 4](https://i.ibb.co/s9HZN10Z/failure-analysis.png) to quantitatively analyze success/failure modes. The results show FlashResearch excels at *"knowledge-intensive"* and *"reasoning-intensive"* queries, which validates our design goals. Meanwhile, the handling of *"ambiguous", "incomplete"* queries is identified as a limitation in our orchestration.

### **d) Methodological Clarifications** (Revisions highlighted in blue color)

* **Expanded Related Work:** We expanded Section 2 to discuss recent multi-agent frameworks like OWL, AgentOrchestra, highlighting FlashResearch’s unique position on **time-efficient**, tree-structured optimization for deep research tasks.
* **Transparency in Orchestration:** We revised Appendix Section B with a more detailed case analysis showing that our use of "thinking models" provides explicit reasoning traces for orchestration decisions, enhancing transparency.

We believe these revisions adequately address the reviewers' concerns. We are grateful for the AC’s efforts and for the reviewers’ constructive feedback. Their insights have greatly improved the quality of our paper and increased our confidence that this work will bring valuable contributions to the broader community.

---

### Meta-Review · Area_Chair_K6Jj · 2025-12-24

**Summary:**

The paper introduces FlashResearch, a framework for efficient deep research that transforms sequential processing into parallel, runtime orchestration by dynamically decomposing complex queries into tree-structured sub-tasks. An adaptive planning module determines the appropriate breadth and depth of exploration, while a real-time orchestration layer monitors progress and optimizes research pathways for maximum efficiency. Experiments show FlashResearch achieves up to a 5x speedup while maintaining or improving report quality under fixed time constraints.

The strengths include 1) the proposed method is logically sound and empirically effective; 2) good speedup by the proposed method; 3) paper is well written. However, the reviewer concerns include:
1. The evaluation is not complete, with only two benchmarks and subset of DeepResearchGym (ZoH6);
2. The use of LLM as a judge to decide when to expand, deepen, or terminate paths, and for evaluation, which may introduce bias (ZoH6, Yz8E, S2LH);
3. The baseline model is only GPT-Researcher, which may increase the concern on generalizability (ZoH6, Yz8E);
4. Missing critical ablations and analysis to further understand the proposed method (ZoH6, YF1V);
5. Extra overhead to call LLM (S2LH);
6. Missing sensitivity analysis across time budgets (S2LH);
7. FlashResearch appears to focus primarily on an offline setting, and tool selection and allocation within online web search scenarios are unclear (YF1V);
8. Missing literature and discussion (YF1V).

Although some of them were addressed by the rebuttal, the AC thinks #1, #2, #4 and #7 are not fully addressed yet. For example:
- Although the authors claimed the 100 samples subset is representative for the full set, the full evaluation can still provide more robust conclusion/observations;
- It is understandable that LLM is being used as judge for DeepResearch works. However, to mitigate the bias, the authors should run different judges for the same benchmarks, instead of different judges for different benchmarks. And running the same judge multiple times doesn’t resolve the bias concern. And if possible, some rule-based judge/evaluation should also be investigated.
- There are benchmarks supporting time-varying tasks evaluation, e.g. Mind2Web 2. The authors claimed the model is able to do online evaluation, but without experiments in such setting.

Given these, the AC thinks there are still critical concern unresolved and this paper is not ready to be published yet. The authors are strongly suggested to revise the paper following the reviewers' feedback.

**Reviewer Concerns:**

The reviewer concerns include:
1. The evaluation is not complete, with only two benchmarks and subset of DeepResearchGym (ZoH6);
2. The use of LLM as a judge to decide when to expand, deepen, or terminate paths, and for evaluation, which may introduce bias (ZoH6, Yz8E, S2LH);
3. The baseline model is only GPT-Researcher, which may increase the concern on generalizability (ZoH6, Yz8E);
4. Missing critical ablations and analysis to further understand the proposed method (ZoH6, YF1V);
5. Extra overhead to call LLM (S2LH);
6. Missing sensitivity analysis across time budgets (S2LH);
7. FlashResearch appears to focus primarily on an offline setting, and tool selection and allocation within online web search scenarios are unclear (YF1V);
8. Missing literature and discussion (YF1V).

Although some of them were addressed by the rebuttal, the AC thinks #1, #2, #4 and #7 are not fully addressed yet.

**Reviewer Scores:**

The pre-rebuttal scores are 4 (ZoH6), 6 (Yz8E), 4 (S2LH), 6 (YF1V). Given the mentioned concerns #1, #2, #4 and #7 are not well addressed, the AC doesn't think ZoH6 and S2LH will raise their scores.

---

### Decision · Program_Chairs · 2026-01-26

Reject